# Colorful Pinball: Density-Weighted Quantile Regression for Conditional Guarantee of Conformal Prediction

**Qianyi Chen**[1]  **Bo Li**[1]

## Abstract

Although conformal prediction provides robust marginal coverage guarantees, achieving reliable conditional coverage for specific inputs remains challenging. While exact distribution-free conditional coverage is impossible with finite samples, recent work has focused on improving the conditional coverage of standard conformal procedures. Distinct from approaches that target relaxed notions of conditional coverage, we directly target the mean squared error of conditional coverage by refining the quantile regression components that underpin many conformal methods. Leveraging a Taylor expansion, we derive a sharp surrogate objective for quantile regression: a density-weighted pinball loss, where the weights are given by the conditional density of the nonconformity score evaluated at the true quantile. We propose a three-headed quantile network that estimates these weights via finite differences using auxiliary quantile levels at $1 - \alpha \pm \delta$, subsequently fine-tuning the central quantile by optimizing the weighted loss. We provide a theoretical analysis with exact non-asymptotic guarantees characterizing the resulting excess risk. Extensive experiments on diverse high-dimensional real-world datasets demonstrate remarkable improvements in conditional coverage performance. We release the code at `CPCP Github repo`.

## 1. Introduction

As machine learning systems are increasingly deployed in high-stakes environments, the demand for reliability extends beyond predictive accuracy to rigorous uncertainty quantification (UQ). In safety-critical domains, a model's ability to provide appropriate predictive intervals is often as valuable

as accurate point predictions. Among existing UQ frameworks, Conformal Prediction (CP; Vovk et al., 2005; Shafer & Vovk, 2008) has emerged as a paradigm of choice due to its mathematically grounded guarantees. Unlike Bayesian methods or ensemble techniques that often rely on strong distributional assumptions or heavy computational overhead, CP offers a distribution-free, model-agnostic framework. It constructs prediction sets or intervals that provably contain the ground truth with a user-specified probability $1 - \alpha$ (e.g., 90%) in finite samples, providing a layer of statistical trust that is indispensable for real-world deployment.

However, standard split conformal prediction only provides a marginal guarantee over the population and cannot guarantee conditional coverage for specific instances—precisely what practitioners require in high-stakes scenarios. Although hardness results for exact distribution-free conditional guarantees with finite samples are well-established (Vovk, 2012; Lei & Wasserman, 2014; Foygel Barber et al., 2021), a growing body of works targets improving the conditional coverage of conformal procedures. To this end, many works seek to relax the conditional requirement (Angelopoulos et al., 2024), e.g., group-conditional coverage (Jung et al., 2023; Ding et al., 2023).

Instead, this paper targets controlling the exact conditional coverage rather than relaxed forms. A natural objective is the MSE of conditional coverage—termed Mean Squared Conditional Error (MSCE) in Kiyani et al. (2024). We consider this metric since minimizing the MSCE provides high-probability control over conditional coverage deviations via Bernstein's inequality. We build upon the connection between MSCE and the excess risk of the pinball loss established in recent literature (Kiyani et al., 2024; Plassier et al., 2025a). Specifically, these works show that the MSCE is upper-bounded by the excess risk of a quantile estimator with pinball loss, suggesting that improving the quantile estimator directly translates to better conditional coverage.

However, we identify that this standard upper bound is often loose, as it relies on a uniform Lipschitz constant for the conditional cumulative distribution function (CDF) of the nonconformity scores, $F_{S|X}(s)$. Using a Taylor expansion, we derive a significantly sharper approximation of MSCE: the excess risk of the pinball loss weighted by

[1]School of Economics and Management, Tsinghua University, China. Correspondence to: Bo Li <libo@sem.tsinghua.edu.cn>.

*Proceedings of the 43$^{rd}$ International Conference on Machine Learning*, Seoul, South Korea. PMLR 306, 2026. Copyright 2026 by the author(s).

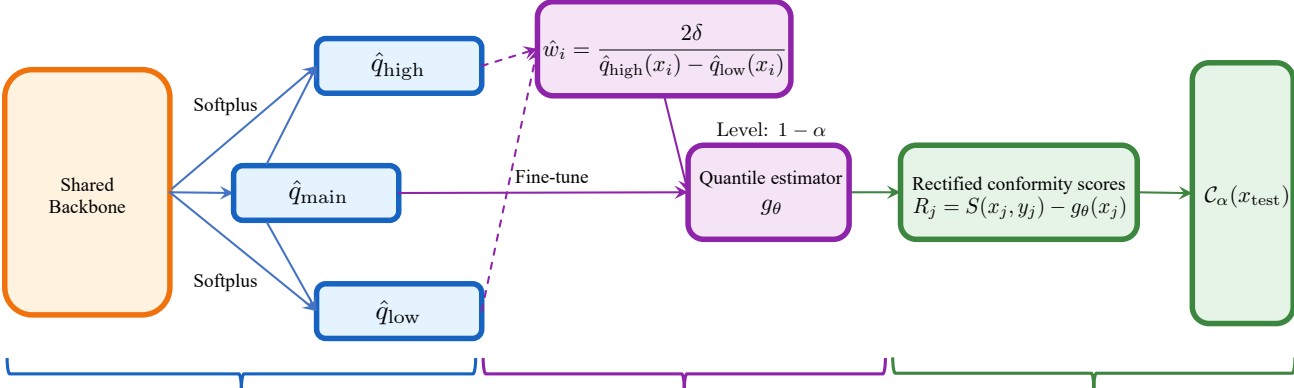

*Figure 1.* **Illustration of the Colorful Pinball Conformal Prediction framework.** CPCP first estimates finite-difference density weights from auxiliary quantiles of the nonconformity-score distribution, then uses these weights to turn the standard pinball loss into a density-weighted objective for fine-tuning the target quantile. This reweighting emphasizes regions where the conditional score distribution is steep at the target quantile, precisely where small quantile errors induce large conditional-coverage errors. The auxiliary quantiles $\hat{q}_{high}$ and $\hat{q}_{low}$ are constructed by adding and subtracting Softplus-activated gap estimates from the central quantile $\hat{q}_{main}$, preventing quantile crossing and ensuring nonnegative finite-difference weights before the final rectified-score conformalization step.

the conditional density of the score evaluated at the true quantile, $f_{S|X}(q(x))$. This formulation reveals the latent heteroscedastic structure within the objective; for instance, in the location-scale family, this weighting term is proportional to the inverse scale, $1/\sigma(x)$.

Motivated by this insight, we propose a three-stage framework named **Colorful Pinball Conformal Prediction (CPCP)**, as shown in Figure 1. Our method first estimates the density-based weights via joint quantile regression, then fine-tunes the target quantile by minimizing the weighted pinball loss, and finally rectifies the original nonconformity scores using the fine-tuned quantile estimator. Theoretically, we establish exact **non-asymptotic** results for the generalization error of the weighted risk, which is of broader interest in problems involving estimated reciprocal weights. To ensure robustness, our algorithm incorporates specific mechanisms to mitigate practical issues such as quantile crossing, unstable inverse weights, and Taylor approximation errors, enabling superior performance on extensive benchmarks. In brief, our main contributions are fourfold:

- We identify the inadequacy of naive quantile regression regarding conditional coverage and construct a significantly sharper approximation of the MSCE.

- We propose a principled algorithmic framework to directly target the MSCE with carefully designed mechanisms to ensure practical robustness.

- We develop a non-asymptotic theory on generalization error with estimated reciprocal weights.

- We validate our approach through comprehensive experiments with extensive ablation studies.

## 1.1. Related Works

We focus our discussion on the conditional coverage of conformal prediction. In the literature, there is another branch of work concerning the *training-conditional* guarantee (Park et al., 2020; Bian & Barber, 2023; Duchi, 2025), which studies the coverage probability conditional on the specific calibration set observed. To clarify, we are specifically concerned with the coverage guarantee conditional on the covariate of the test sample, $X_{test}$.

Formally, exact conditional coverage refers to $\mathbb{P}(Y_{test} \in \mathcal{C}_\alpha(X_{test}) \mid X_{test})$, which is the ideal quantity we seek to control. In contrast, relaxed versions target quantities such as $\mathbb{P}(Y \in \mathcal{C}_\alpha(X_{test}) \mid H(X_{test}))$ for a certain function $H$.

Group-conditional coverage serves as a natural approximation to the exact conditional guarantee. Here, we specifically discuss the case where groups are formed by partitioning the covariate space. Classic conformal procedures operate with predefined groups (Jung et al., 2023), where $h$ is a fixed discrete partition function. Beyond fixed groups, Kiyani et al. (2024) propose co-training the partition function with quantile regression within each group, providing a discrete approximation of conditional coverage. Though straightforward, the precision of this discrete approximation is limited, and the convergence rate of the resulting MSCE scales no faster than $O(n^{-1/4})$. Adopting a different perspective that reformulates the conditional guarantee as moment equations, Gibbs et al. (2025a) extend the approximation to infinite, overlapping groups. They leverage regularized quantile regression as well as full conformal prediction (Vovk et al., 2005). However, Gibbs et al. (2025a) provides control only on a substitute metric—coverage under covariate shift—

rather than the exact conditional coverage. Moreover, full conformal procedures incur a significant computational burden, limiting their general applicability.

Beyond group-conditional guarantees, localized conformal prediction (Bian & Barber, 2023; Hore & Barber, 2025) provides another avenue to approximate the conditional guarantee by applying kernel weighting during empirical quantile computation. As analyzed in Hore & Barber (2025), the guarantee provided by such localization corresponds to a random function $h$ that outputs a point sampled from a distribution centered at $X_{\text{test}}$, defined by the kernel. However, this strategy suffers inherently from the curse of dimensionality, particularly in the complex, high-dimensional scenarios that modern conformal prediction targets.

Complementing the focus on grouping and quantile computation, a substantial body of literature strives to refine the nonconformity score function. Improving upon the classic residual score, Lei et al. (2018) propose scaled residuals using a learned conditional standard deviation $\sigma(x)$. Romano et al. (2019) propose Conformalized Quantile Regression (CQR), which directly conformalizes two learned quantiles rather than a mean prediction. They also highlight the difficulty of learning $\sigma(x)$, particularly for overfitted neural networks. More recently, Xie et al. (2024) propose a boosting procedure to iteratively refine the score function. However, in addition to high computational costs, this method further requires access to training data, which is often infeasible given the prevalence of pre-trained black-box models. Similarly, CQR necessitates replacing the traditional conditional mean regression objective with quantile regression on the training set, thereby limiting its general applicability. While the aforementioned scores are tailored for one-dimensional labels, another branch of work investigates density-based scores (Izbicki et al., 2022; Plassier et al., 2025b; Braun et al., 2026), which are readily extensible to multi-dimensional labels and also flexible enough to capture heteroscedasticity. However, estimating the conditional density $f(Y \mid X)$ is generally a harder task than regression on the conditional mean $\mathbb{E}[Y \mid X]$. Furthermore, the calibration set used for density estimation is typically much smaller than the training set. This sample size mismatch limits the efficacy of density estimation; therefore, we focus on the quantile regression of scores rather than density estimation.

Most relevant to our work is the approach of performing quantile regression directly on conformity scores (Gibbs et al., 2025a; Kiyani et al., 2024; Plassier et al., 2025a). As mentioned before, the excess risk of the pinball loss constitutes an upper bound on the MSCE. Specifically, Plassier et al. (2025a) propose a general framework called Rectified Conformal Prediction (RCP), which transforms the original nonconformity score into a rectified version that tries

to remove the covariate-dependent component of the score and thereby naturally improves conditional coverage of standard split conformal prediction with these rectified scores. However, naive quantile regression using the pinball loss exhibits inherent limitations in conditional coverage (Feldman et al., 2021), and struggles even with marginal coverage in high-dimensional settings (Gibbs et al., 2025b). In our context, we identify that naive quantile regression overlooks a critical heteroscedastic component. Our method aims to recover this missing component, yielding a significantly sharper approximation to the MSCE.

## 2. Preliminaries

**Split Conformal Prediction.** We consider the standard setting where we observe i.i.d. data points $(X, Y) \in \mathcal{X} \times \mathcal{Y}$. In the split conformal prediction framework (Papadopoulos et al., 2002), the available data are randomly partitioned into two disjoint subsets: a training set $\mathcal{D}_{\text{train}}$ and a calibration set $\mathcal{D}_{\text{cal}} = \{(X_i, Y_i)\}_{i=1}^n$ of size $n$. A predictive model is first fitted on $\mathcal{D}_{\text{train}}$. We then define a nonconformity score function $S : \mathcal{X} \times \mathcal{Y} \to \mathbb{R}$, which measures the discrepancy between the target $y$ and the model's prediction at $x$ (e.g., the absolute residual $S(x, y) = |y - \hat{\mu}(x)|$). The scores are computed for all calibration points as $S_i = S(X_i, Y_i)$ for all $i \in \{1, \ldots, n\}$. Given a user-specified miscoverage level $\alpha \in (0, 1)$, we compute the conformal threshold $\hat{q}$ as the $\lceil (n + 1)(1 - \alpha) \rceil / n$-th empirical quantile of the calibration scores $\{s_1, \ldots, s_n\}$. For a new test input $X_{n+1}$, the prediction set is constructed as $\mathcal{C}(X_{n+1}) = \{y \in \mathcal{Y} : S(X_{n+1}, y) \leq \hat{q}\}$. This procedure satisfies:

$$1 - \alpha \leq \mathbb{P}(Y_{n+1} \in \mathcal{C}(X_{n+1})) \leq 1 - \alpha + \frac{1}{n + 1}. \quad (1)$$

**Rectified Conformal Prediction (RCP).** As has been widely noted, the failure pattern of standard split conformal prediction is undercoverage in hard regions and overcoverage in easy ones. To improve the conditional coverage, Plassier et al. (2025a) propose RCP, which transforms the raw nonconformity scores into a rectified version that is approximately homoscedastic. While the RCP framework allows for general transformations, we focus on the fundamental additive correction in our work.

Specifically, let $\hat{q}_{1-\alpha} : \mathcal{X} \to \mathbb{R}$ be an estimator of the conditional $(1 - \alpha)$-quantile of the raw score $S(X, Y)$ given $X$. The rectified score function $R : \mathcal{X} \times \mathcal{Y} \to \mathbb{R}$ is defined as the deviation from this estimated quantile:

$$R(x, y) := S(x, y) - \hat{q}_{1-\alpha}(x). \quad (2)$$

The conformal procedure is then applied to these rectified scores. We compute the rectified scores on the calibration set, $\mathcal{R}_{\text{cal}} = \{R(x_i, y_i)\}_{i=1}^n$, and find their $\lceil (n + 1)(1 - $

$\alpha)\rceil/n$-th empirical quantile, denoted by $\hat{\gamma}$. The resulting prediction set for a new test point $X_{n+1}$ is constructed as:

$$\mathcal{C}(X_{n+1}) = \{y \in \mathcal{Y} : S(X_{n+1}, y) \le \hat{q}_{1-\alpha}(X_{n+1}) + \hat{\gamma}\}. \tag{3}$$

Intuitively, $\hat{q}_{1-\alpha}(x)$ serves as a coarse, instance-dependent baseline threshold, while $\hat{\gamma}$ acts as a global, residual correction to ensure exact marginal validity.

**Conditional coverage.** Given marginal validity, our primary concern is the *conditional coverage*, defined as:

$$\pi(x) := \mathbb{P}(Y \in \mathcal{C}_\alpha(X) \mid X = x), \tag{4}$$

which characterizes the coverage probability for each specific instance $x$. Rather than considering any relaxed form, we focus on controlling the exact conditional coverage.

Since $\pi(x)$ is a function of $x$, a natural approach is to construct a metric that summarizes its deviation from the target level $1 - \alpha$. Following Kiyani et al. (2024), we define the Mean Squared Coverage Error (MSCE) as:

$$\mathrm{MSCE} := \mathbb{E}[(\pi(X) - (1 - \alpha))^2]. \tag{5}$$

We further motivate this metric through a novel lens. Viewing $\pi(X)$ as a random variable, the tower property implies that its expectation is the marginal coverage:

$$\mathbb{E}[\pi(X)] = \mathbb{P}(Y \in \mathcal{C}_\alpha(X)), \tag{6}$$

which is controlled by marginal validity as stated in Equation 1. Given this, achieving conditional validity reduces to establishing concentration of $\pi(X)$, which naturally amounts to controlling its variance. Marginal validity ensures that $\mathbb{E}[\pi(X)]$ converges to the target level $1 - \alpha$ at the fast rate $O(n^{-1})$; consequently, the bias term is negligible, and the variance is nearly equivalent to the MSCE.

**Quantile regression.** Our framework relies on estimating the conditional quantiles of the nonconformity scores. Formally, let $Z \in \mathbb{R}$ be a target random variable (typically the score $S$ in our context) and $X \in \mathcal{X}$ be the covariates. We denote the true conditional $\tau$-th quantile of $Z$ given $X = x$ as $q_\tau(x) := \inf\{z : \mathbb{P}(Z \le z \mid X = x) \ge \tau\}$ for a level $\tau \in (0, 1)$. Standard quantile regression estimates $q_\tau$ by minimizing the expected pinball loss (Koenker, 2005). The pinball loss function $\rho_\tau : \mathbb{R} \times \mathbb{R} \to \mathbb{R}_{\ge 0}$ is defined as:

$$\rho_\tau(q, u) := \max\{\tau(u - q), (\tau - 1)(u - q)\}. \tag{7}$$

In practice, given a hypothesis class $\mathcal{G}$ (e.g., neural networks), we seek an estimator $\hat{g} \in \mathcal{G}$ that minimizes the empirical risk $\frac{1}{n}\sum_{i=1}^n \rho_\tau(z_i, g(x_i))$. A fundamental property of this objective is that, assuming sufficient model capacity, the population minimizer $g^\star := \arg\min_g \mathbb{E}[\rho_\tau(Z, g(X))]$ uniquely recovers the true conditional quantile $q_\tau$.

## 3. Approximation to MSCE

In general, without a generative model, we typically observe only a single realization of $Y$ for each instance $X$. As a result, although the MSCE is a theoretically well-motivated metric, it is inherently difficult to evaluate—and hence to optimize—in practice. To address this challenge, Kiyani et al. (2024) derived an upper bound on the MSCE, while Plassier et al. (2025a) established a stronger pointwise bound on the deviation $|F_{S|X}(\hat{q}_{1-\alpha}(x)) - (1 - \alpha)|$. Though there is a gap between $F_{S|X}(\hat{q}_\tau(x))$ and $\pi(x)$ in RCP due to the conformalization step after quantile regression, Plassier et al. (2025a) proves that it shrinks at an exponential rate with respect to the size of sample used for conformalization.

Hereafter, we will use $\tau := 1 - \alpha$ to denote the target coverage level, and $\hat{q}_\tau(x)$ to denote the estimated conditional $\tau$-quantile of the nonconformity scores. We now introduce the pointwise results that motivate quantile regression with standard pinball loss (Plassier et al., 2025a).

**Proposition 3.1.** *If the conditional CDF of scores is $L_F$-Lipschitz, then under mild regularity conditions,*

$$|F_{S|X}(\hat{q}_\tau(x)) - \tau| \le \sqrt{2L_F(\mathcal{L}_x(\hat{q}_\tau(x)) - \mathcal{L}_x(q_\tau(x)))} \tag{8}$$

*holds. Here, $\mathcal{L}_x(\cdot) = \mathbb{E}_Y[\rho_\tau(\cdot, s(x, Y))]$ denotes the expected pinball loss (with expectation taken over $Y \mid X$).*

The proof is detailed in Appendix B.1. Next, we introduce the assumption on consistency of quantile estimators and our approximation for the MSCE based on Taylor expansion.

**Assumption 3.2.** *We assume the quantile estimator $\hat{q}_\tau$ is $L_2$-consistent, i.e., $\epsilon_q(x) := \hat{q}_\tau(x) - q_\tau(x)$ satisfies:*

$$\|\epsilon_q\|_{L_2(\mathbb{P}_X)} \xrightarrow{p} 0 \quad \text{as } n \to \infty \tag{9}$$

*Remark* 3.3. Assumption 3.2 requires only $L_2$-consistency to identify the leading term in Proposition 3.4. In later sections, when analyzing the finite-sample performance, we establish a fast convergence rate for $\hat{q}_\tau$.

We then define the Mean Squared Quantile Error (MSQE) as the first step to derive a tractable surrogate of MSCE.

$$\mathrm{MSQE}\,(\hat{q}_\tau) := \mathbb{E}_X\left[\left(F_{S|X}(\hat{q}_\tau(x)) - \tau\right)^2\right]. \tag{10}$$

Next, to circumvent estimating the full conditional distribution $F_{S|X}$, we expand around $q_\tau$, which constitutes the key step toward tractability. This is achieved via Taylor expansions, as formalized in the following proposition.

**Proposition 3.4.** *Let $G(u) := \left(F_{S|X}(u) - \tau\right)^2$, and let $\mathcal{E}(x) := \mathcal{L}_x(\hat{q}_\tau(x)) - \mathcal{L}_x(q_\tau(x))$ denote the pointwise excess risk under pinball loss. Under standard regularity assumptions, the following two Taylor expansions hold:*

$$G(\hat{q}_\tau(x)) = f_{S|X}(q_\tau(x))^2\epsilon_q(x)^2 + \frac{1}{6}G'''(\xi_{S,1})\,\epsilon_q(x)^3, \tag{11}$$

*and*

$$\mathcal{E}(x) = \frac{1}{2}f_{S|X}(q_\tau(x))\epsilon_q(x)^2 + \frac{1}{6}f'_{S|X}(\xi_{S,2})\epsilon_q(x)^3, \quad (12)$$

*where $\xi_{S,1}$ and $\xi_{S,2}$ both lie between $\hat{q}_\tau(x)$ and $q_\tau(x)$.*

*Under Assumption 3.2 and mild regularity conditions on the density $f_{S|X}$, the squared conditional quantile error admits the following expansion:*

$$\left(F_{S|X}(\hat{q}_\tau(x)) - \tau\right)^2 = 2f_{S|X}(q_\tau(x))\mathcal{E}(x) + C_f\epsilon_q(x)^3, \quad (13)$$

*where $C_f$ is a constant that characterizes the smoothness of $f_{S|X}$. Consequently, the MSQE satisfies:*

$$\text{MSQE} = 2\mathbb{E}_X\left[f_{S|X}(q_\tau(X))\mathcal{E}(X)\right] + C_f\mathbb{E}_X[\epsilon_q(X)^3], \quad (14)$$

*and the first term serves as our final optimization surrogate.*

The proof of Proposition 3.4 is provided in Appendix B.2. The key implication is that the standard pinball objective and the conditional-coverage objective penalize quantile errors under different local geometries. Writing $\epsilon_q(x) := \hat{q}_\tau(x) - q_\tau(x)$, Proposition 3.4 shows that the leading term of the squared coverage deviation scales as $f_{S|X}(q_\tau(x))^2\epsilon_q(x)^2$, whereas the leading term of the pointwise pinball excess risk scales as $f_{S|X}(q_\tau(x))\epsilon_q(x)^2$. Thus, plain pinball loss underweights regions where the conditional CDF is steep at the target quantile, precisely where a small quantile error can induce a large coverage error. Multiplying the pinball loss by the oracle density weight $f_{S|X}(q_\tau(x))$ aligns its leading quadratic term with the MSQE up to an irrelevant constant factor. To make this density weight more interpretable, we next consider the location-scale family as an example.

**Example 3.1.** *Consider the case where the nonconformity score $S$ given $X$ follows a location-scale family:*

$$S = \varpi(x) + \sigma(x)\xi, \quad (15)$$

*where $\xi$ is a standardized random variable with a base PDF $f_0(\cdot)$ (e.g., standard normal or Laplace) and CDF $F_0(\cdot)$. The conditional density of $S$ is given by $f_{S|X}(s) = \frac{1}{\sigma(x)}f_0\left(\frac{s-\varpi(x)}{\sigma(x)}\right)$. Let $z_\tau = F_0^{-1}(\tau)$ be the $\tau$-quantile of the base distribution. The true conditional quantile of the score is then $q_\tau(x) = \varpi(x) + \sigma(x)z_\tau$. Evaluating the conditional density at the true quantile yields:*

$$f_{S|X}(q_\tau(x)) = \frac{1}{\sigma(x)}f_0(z_\tau) \propto \frac{1}{\sigma(x)}. \quad (16)$$

This example shows that minimizing the standard pinball loss collapses the spectrum of heteroscedasticity inherent in the MSQE objective into a single, unweighted error criterion. Motivated by the need to recover this heteroscedastic

spectrum, we refer to our approach as colorful pinball, in contrast to the conventional plain pinball loss.

The severity of this objective misalignment depends on the degree of heteroscedasticity in the scores (and intrinsically, in $Y \mid X$). We note that standard quantile regression already captures part of the heteroscedasticity, yet our approximation reveals that **additional emphasis on heteroscedasticity** is needed when targeting conditional coverage. To see this, recall Equation (11) and (12). Therefore, although minimizing either the standard or density-weighted pinball loss recovers the true quantile asymptotically, a severe misalignment persists with finite samples, which is precisely the misalignment our weighting scheme is designed to correct.

*Remark* 3.5. In Example 3.1, the weight $1/\sigma(x)$ may appear counterintuitive as it assigns lower importance to high-variance regions. However, we note that this weight is put on the distance of two quantiles, and a small $\sigma(x)$ corresponds to a steeper $F_{S|X}$, making the coverage probability highly sensitive to estimation errors; a slight deviation in $\hat{q}_\tau(x)$ can drastically degrade coverage (e.g., from 95% to 80%). Thus, normalizing by $\sigma(x)$ effectively standardizes this sensitivity across instances, promoting a stable coverage independent of $x$, i.e., $\pi(x) \approx \text{const}$.

# 4. Colorful Pinball Conformal Prediction

## 4.1. Algorithm Details

Figure 1 illustrates the proposed workflow. We begin by partitioning the calibration set $\mathcal{D}_{\text{cal}}$ into three disjoint subsets: $\mathcal{D}_{\text{cal},1}, \mathcal{D}_{\text{cal},2}$, and $\mathcal{D}_{\text{cal},3}$. Our objective targets the leading term in Equation (14), which can be reformulated as:

$$\begin{aligned}&\mathbb{E}_X[f_{S|X}(q_\tau(X))\mathcal{E}(X)]\\&= \mathbb{E}_{X,Y}\left[f_{S|X}(q_\tau(X))\rho_\tau(\hat{q}_\tau(X), S)\right] + \text{const}\end{aligned} \quad (17)$$

To optimize this empirically, we require an estimate of the weight $w(x) := f_{S|X}(q_\tau(X))$. Instead of estimating the full conditional density $f_{S|X}$, we note that the quantile function is the inverse function of the CDF. Thus, we have:

$$\frac{\partial q_\tau(x)}{\partial \tau} = \frac{1}{f_{S|X}(q_\tau(x))}. \quad (18)$$

Consequently, we approximate the density weight using the finite-difference estimator:

$$\hat{w}(x) = \frac{2\delta}{\hat{q}_{\tau+\delta}(x) - \hat{q}_{\tau-\delta}(x)}. \quad (19)$$

This offers a clear benefit: it requires estimating only two auxiliary quantiles, $q_{\tau+\delta}(x)$ and $q_{\tau-\delta}(x)$, alongside the primary target $q_\tau(x)$. This structure naturally motivates a multitask learning framework. We employ a shared feature

extractor $h(x)$ coupled with three distinct output heads:

$$\hat{q}_\tau(x) = \phi_{\text{main}} \circ h(x)$$
$$\hat{q}_{\tau+\delta}(x) = \hat{q}_\tau(x) + \text{Softplus}(\phi_{\text{high}} \circ h(x)) \qquad (20)$$
$$\hat{q}_{\tau-\delta}(x) = \hat{q}_\tau(x) - \text{Softplus}(\phi_{\text{low}} \circ h(x)).$$

Here, the $\text{Softplus}(\cdot) = \log(1 + \exp(\cdot))$ activation is employed to ensure monotonicity, preventing quantile crossing that would otherwise yield invalid negative weights $w_i$. In practice, $h$ and the projection heads $\{\phi_\cdot\}$ are parameterized by neural networks (e.g., MLPs).

We observe that the approximation error derived from the Taylor expansion in Equation (14) is contingent on the accuracy of the quantile estimates. Therefore, we first perform joint training of all three estimators on $\mathcal{D}_{\text{cal},1}$ using the standard pinball loss. With this initialization, we freeze the backbone $h$ and fine-tune the primary head $\phi_{\text{main}}$ on $\mathcal{D}_{\text{cal},2}$:

$$\phi_{\text{main}} = \underset{\phi}{\arg\min} \sum_{i \in \mathcal{D}_{\text{cal},2}} \hat{w}(x_i) \rho_\tau(\phi \circ h(x_i), s_i). \qquad (21)$$

The resulting quantile estimator is the final version: $\hat{q}_\tau = \phi_{\text{main}} \circ h$. To ensure marginal validity, we apply RCP by computing the residuals on $\mathcal{D}_{\text{cal},3}$:

$$R_j = S_j - \hat{q}_\tau(x_j). \qquad (22)$$

We then compute $\hat{\gamma}$ as the $\lceil(|\mathcal{D}_{\text{cal},3}| + 1)(\tau)\rceil$-th smallest value of $\{R_j\}_{j \in \mathcal{D}_{\text{cal},3}}$ and construct the conformal set:

$$\mathcal{C}_\alpha(x_{\text{test}}) = \{y : S(x_{\text{test}}, y) \le \hat{\gamma} + \hat{q}_\tau(x_{\text{test}})\}. \qquad (23)$$

For instance, the classic absolute residual score $S(x, y) = |y - \hat{\mu}(x)|$ for scalar regression yields:

$$\mathcal{C}_\alpha(x_{\text{test}}) = [\hat{\mu}(x_{\text{test}}) \pm (\hat{q}_\tau(x_{\text{test}}) + \hat{\gamma})]. \qquad (24)$$

The complete procedure is summarized in Algorithm 1.

## 4.2. Towards Better Finite-Sample Stability

We now introduce supplementary mechanisms to enhance the empirical stability of our methodology. As defined in Equation (19), the proposed estimator relies on inverse weighting, where the denominator corresponds to the estimated inter-quantile gap $\hat{q}_{\tau+\delta}(x_i) - \hat{q}_{\tau-\delta}(x_i)$. A vanishing denominator—often caused by estimation errors—can yield arbitrarily large weights $w_i$, leading to explosive variance in the optimization objective. Theoretically, ensuring stability requires the sample size $n$ to be sufficiently large for the neural network to reliably distinguish the $(\tau - \delta)$ and $(\tau + \delta)$ quantiles (e.g., 89% vs. 91%), thereby keeping the estimated gap bounded away from zero.

To stabilize the fine-tuning procedure, particularly in regimes where the bandwidth $\delta$ is small relative to $n$, we propose two strategies: *weight clipping* and *loss mixing*.

---

**Algorithm 1** Colorful Pinball Conformal Prediction

1: **Input:** Calibration data $\mathcal{D}_{\text{cal}}$, Point predictor $\hat{\mu}$, Test input $x_{\text{test}}$, Target coverage level $\tau$, Bandwidth $\delta$
2: Split $\mathcal{D}_{\text{cal}}$ into three disjoint parts $\mathcal{D}_{\text{cal},1}, \mathcal{D}_{\text{cal},2}, \mathcal{D}_{\text{cal},3}$.
3: Compute conformity scores $S_i$ for all $(x_i, y_i) \in \mathcal{D}_{\text{cal}}$.
4: Train three quantile estimators $\hat{q}_{\text{low}}, \hat{q}_{\text{main}}$, and $\hat{q}_{\text{high}}$ (for levels $\tau - \delta, \tau, \tau + \delta$) jointly on $\mathcal{D}_{\text{cal},1}$.
5: Compute the weight for all $(x_i, y_i) \in \mathcal{D}_{\text{cal},2}$:

$$w_i \leftarrow \frac{2\delta}{\hat{q}_{\text{high}}(x_i) - \hat{q}_{\text{low}}(x_i)}.$$

6: Fine-tune $\hat{q}_{\text{main}}$ on $\mathcal{D}_{\text{cal},2}$ to get the final quantile estimator $\hat{q}_\tau$ (freezing backbone and auxiliary heads).
7: Compute residuals for all $(x_j, y_j) \in \mathcal{D}_{\text{cal},3}$:

$$R_j \leftarrow S_j - \hat{q}_\tau(x_j)$$

8: Compute empirical quantile $\hat{\gamma}$ as the $\lceil(|\mathcal{D}_{\text{cal},3}|+1)\tau\rceil$-th smallest value of $\{R_j\}$.
9: **Output:** $\mathcal{C}_\alpha(x_{\text{test}}) = \{y : S(x_{\text{test}}, y) \le \hat{\gamma} + \hat{q}_\tau(x_{\text{test}})\}$

---

- **Weight Clipping:** We truncate excessive weights to a threshold. This threshold can be set as a multiple $M$ of the empirical mean of the weights.

- **Loss Mixing:** We modify the optimization objective as a convex combination of the (self-normalized) weighted pinball loss and the standard pinball loss.

In essence, these two strategies correspond to imposing artificial upper and lower bounds on the estimated weights, a modification motivated by the theoretical bottlenecks identified in the proof of our main theorem (Theorem 5.2). Furthermore, such clipping strategies are widely recognized as effective in settings involving reciprocal weighting; they significantly reduce variance at the cost of introducing slight bias, a trade-off commonly utilized in policy evaluation (e.g., clipping inverse propensity weights; Swaminathan & Joachims, 2015). We demonstrate the empirical benefits of these strategies in Section 6 and Appendix A.5.

## 5. Theoretical Analysis

To approximate MSCE, i.e., the MSE of coverage probability $\pi(x)$, we employ a two-step transformation. First, we substitute the target $\pi(x)$ with $F_{S|X}(\hat{q}_\tau(x))$; second, we select the leading term derived in Proposition 3.4 as our final optimization surrogate. According to Algorithm 1, the first substitution introduces a discrepancy due to the global conformal shift $\hat{\gamma}$ required for marginal coverage. However, as shown by Plassier et al. (2025a), this gap decays

exponentially with the calibration sample size $m$[1], provided $\hat{q}_\tau$ satisfies certain regularity conditions that control the quality of the quantile estimator $\hat{q}_\tau$ (see Appendix C.1). Intuitively, with a moderately large sample size of $\mathcal{D}_{\text{cal}}$ (e.g., in the hundreds), we can focus on characterizing the finite-sample performance of our method by analyzing the MSE of $F_{S|X}(\hat{q}_\tau(x))$, and, consequently, the expected risk of the density-weighted pinball surrogate.

Our theoretical goal is therefore to establish a finite-sample generalization guarantee for this density-weighted pinball objective. This is an appropriate target for two reasons. First, Proposition 3.4 shows that the excess density-weighted pinball risk is the leading tractable surrogate for the MSQE, up to higher-order Taylor remainders. Thus, controlling this excess risk translates into controlling the squared deviation $F_{S|X}(\hat{q}_\tau(X)) - \tau$ in expectation. Second, the remaining gap between this quantity and the final conditional coverage $\pi(X)$ is introduced only by the conformalization shift, whose effect is exponentially small in the size of the final calibration split under the regularity conditions discussed in Appendix C.1. Consequently, a finite-sample excess-risk bound for the weighted objective provides a direct route from the empirical optimization problem solved by CPCP to the desired control of conditional-coverage error.

We now present our main technical assumptions as follows.

**Assumption 5.1.** Let $\mathcal{X} \subset \mathbb{R}^d$ be compact. We assume:

1. **Quantile smoothness in $\tau$.** The conditional quantile function $q_\tau(x)$ is three times continuously differentiable with respect to $\tau$ for all $x \in \mathcal{X}$, and

$$\sup_{x \in \mathcal{X}} \left| \frac{\partial^3 q_\tau(x)}{\partial \tau^3} \right| \le B_q'''.$$

2. **Density regularity at the target quantile.** There exist constants $b_w, B_w$ such that:

$$0 < b_w \le f_{S|X}(q_\tau(x) \mid x) \le B_w < \infty, \quad \forall x \in \mathcal{X},$$

3. **Local Hölder-type norm equivalence.** There exist constants $C_{\text{norm}} > 0$, $\nu \in [1/3, 1]$, and $r > 0$ such that for all $g \in \mathcal{G}$ satisfying $\|g - g^\star\|_{L_2(\mathbb{P}_X)} \le r$,

$$\|g - g^\star\|_{L_\infty(\mathcal{X})} \le C_{\text{norm}} \|g - g^\star\|_{L_2(\mathbb{P}_X)}^\nu.$$

Among these assumptions, the third is less conventional and warrants a more detailed discussion due to its pivotal role in our proof. The primary challenge in bounding the generalization error of our method stems from the estimated reciprocal weights: specifically, we require a pointwise lower bound on the denominator, whereas typical guarantees

---

[1]We denote $|\mathcal{D}_{\text{calib},3}| = m$ and $|\mathcal{D}_{\text{calib},1}| = |\mathcal{D}_{\text{calib},2}| = n$.

yield only $L_2$ convergence. To bridge this gap, we introduce this norm-equivalence assumption, which we require it to hold only locally within a neighborhood of the true quantile function $g^\star$. In general, it suffices that $g - g^\star$ belongs to a Hölder class for this assumption to hold. We provide a proof of this claim in Appendix B.4, which can also be obtained as a special case of the celebrated Gagliardo–Nirenberg interpolation inequality. Finally, we note that the exponent $\nu$ increases with the smoothness of the function class; for instance, $\nu = 1$ for parametric models or regression in a Reproducing Kernel Hilbert Space with a smooth kernel.

We then introduce the necessary definitions for presenting the main theorem. Let $\mathcal{G}$ be a hypothesis class with empirical Rademacher complexity $\mathfrak{R}_n(\mathcal{G})$. Let $\mathcal{R}(g) = \mathbb{E}_X[w(X)\rho_\tau(g(X), S)]$ be the expected risk of weighted pinball loss. Let the weights be estimated by $\hat{w}(x) = 2\delta/(\hat{q}_{\tau+\delta}(x) - \hat{q}_{\tau-\delta}(x))$ where the two $\hat{q}_\beta \in \mathcal{G}$ are the auxiliary quantile estimators derived through empirical risk minimization (ERM) with standard pinball loss. Let $\mathcal{E}_q(n) := \sup_{\beta \in \{\tau-\delta, \tau+\delta\}} \|\hat{q}_\beta - q_\beta\|_{L_2(\mathbb{P}_X)}$ denote the estimation error of the two auxiliary quantiles. Finally, let $M_{\rho,\mathcal{G}} := \sup_{g \in \mathcal{G}}(\mathbb{E}_n[\rho_\tau(g(X), S)^2])^{1/2}$, and let $\sigma_S$ denote the bound on conditional sub-Gaussian scale of scores.

**Theorem 5.2.** *Under Assumption 5.1, with probability at least $1 - 3\zeta$, setting the finite-difference bandwidth to $\delta^\star \asymp (\mathfrak{R}_n(\mathcal{G}))^{1/3}$, with appropriately large $n$, we have:*

$$\mathcal{R}(\hat{g}) - \mathcal{R}(g^\star) = O\left(\mathfrak{R}_n(\mathcal{G})^{\frac{2}{3}}\right), \tag{25}$$

*which is of $O(n^{-1/3})$ when a fast rate is established via local Rademacher complexity.*

*Specifically, the exact non-asymptotic bound is given by:*

$$\begin{aligned}
\mathcal{R}(\hat{g}) - \mathcal{R}(g^\star) &\le C_1 \mathfrak{R}_n(\mathcal{G}) + C_2 \sqrt{\frac{\log(1/\zeta)}{n}} \\
&+ C_3 M_{\rho,\mathcal{G}} \left(\mathcal{E}_q(n)^2 + C_4 \mathcal{E}_q(n)^{1+\nu}\sqrt{\frac{\log(1/\zeta)}{n}}\right. \\
&+ \left. C_5 \frac{\mathcal{E}_q(n)^{2\nu} \log(1/\zeta)}{n}\right)^{1/3},
\end{aligned}$$

$$\tag{26}$$

*where constants $C_1 = 4B_w L_\rho$, $C_2 = 2\sqrt{2}B_w L_\rho \sigma_S$, $C_3 = 4 \times 3^{2/3} B_w^2 (B_q''')^{1/3}$, $C_4 = 2^{\nu-1/2}C_{\text{norm}}$, $C_5 = C_{\text{norm}}^2/6$, $L_\rho = \max\{\tau, 1-\tau\}$.*

Compared to direct quantile regression with the pinball loss, our method exhibits a slower asymptotic rate due to the additional error introduced by estimating density-based weights. However, we emphasize that the primary value of conformal prediction lies in the finite-sample regime, which motivates the exact non-asymptotic expression in our theory. In finite samples, although direct quantile regression may converge faster, the misalignment of its optimization objective with

conditional coverage can substantially limit its performance, as illustrated in Proposition 3.4 and Example 3.1. Moreover, when higher-order central finite-difference schemes of order $k$ are employed (Fornberg, 1988), the resulting excess risk rate can be further improved to $O(\mathfrak{R}_n(\mathcal{G})^{2k/(2k+1)})$.

# 6. Experiments

In this section, we present extensive experiments across eight classic datasets to validate our methodology. Due to space constraints, we defer the details of the data description and detailed experimental results to Appendix A.

**Data.** We consider eight real-world regression benchmarks spanning engineering systems, energy, transportation, and economics. Among them, three datasets (Bike, Diamond, and Superconductivity) have one-dimensional responses, while the remaining five involve multi-dimensional responses. The covariate dimension ranges from a moderate scale (tens of features), and the sample size varies between 10k and 70k. Each dataset is randomly split into training, calibration, and test sets with a ratio of 6:2:2.

**Baselines.** We consider standard baselines including Split Conformal Prediction (Split; Papadopoulos et al., 2002), Partition Learning Conformal Prediction (PLCP; Kiyani et al., 2024), Gaussian Scoring (Braun et al., 2026), CQR (Romano et al., 2019), and RCP (Plassier et al., 2025a). In addition, to separately validate the value of our approximation described in Proposition 3.4, we select a parametric likelihood for the nonconformity score, the Asymmetric Laplace Distribution (ALD), whose maximum likelihood estimation (MLE) is equivalent to minimizing scaled pinball loss and a regularization term for the scale function $\sigma(x)$. We construct two baselines based on CQR and RCP by substituting the quantile regression with MLE of ALD. We provide the details of the baselines in Appendix A.2.

Except for Gaussian Scoring, we adopt $\ell_\infty$ norm of residuals as nonconformity scores (Diquigiovanni et al., 2024), which reduces to the standard absolute residual score for the case of one-dimensional labels.

**Metrics.** In real-world datasets, the data-generating conditional law is unavailable, so pointwise conditional coverage cannot be evaluated directly. We therefore use several complementary diagnostics. First, MSCE, approximated by $K$-means partitions, estimates the squared fluctuation of the conditional coverage function around the target level; under marginal validity, a smaller MSCE indicates that coverage is less driven by averaging over over-covered and under-covered regions. Second, Worst-Slice Coverage (WSC; Cauchois et al., 2021; Romano et al., 2020) reports the lowest empirical coverage over data-dependent slices, and thus serves as a robustness diagnostic for localized undercoverage. Third, we report $L_1$-ERT and $L_2$-ERT, the excess-risk

of the target coverage diagnostics proposed by Braun et al. (2025). These metrics measure whether the coverage indicator is predictable from the covariates: if conditional coverage is well calibrated, an auditor using $X$ should not substantially improve over the constant target-coverage predictor. The $L_1$ version captures average absolute predictable deviations, whereas the $L_2$ version emphasizes larger localized deviations and is closer in spirit to MSCE. Overall, better conditional coverage is indicated by lower MSCE, lower $L_1/L_2$-ERT, and higher WSC. Throughout the experiments, we set the target coverage level to $\tau = 90\%$. To evaluate efficiency, we also report the mean per-dimension log volume of the prediction set.

All methods are evaluated over 20 Monte Carlo repetitions, and we report $\mathrm{mean} \pm \mathrm{std}$. Metric details and complete results are provided in Appendices A.4 and A.5, respectively.

## 6.1. Results Analysis

We present the main conditional-coverage results in Tables 1, 2, 3, and 4; complete results, including ablations and efficiency comparisons, are provided in Appendix A.5. Across all eight benchmarks, CPCP consistently improves conditional-coverage diagnostics over strong baselines. The gains are especially clear for WSC, where higher values indicate fewer severe localized undercoverage failures. Equivalently, the reduction in $\tau - \mathrm{WSC}$ shows that CPCP substantially improves worst-slice miscoverage. The improvements in $L_1$-ERT and $L_2$-ERT further suggest that the remaining coverage errors are less predictable from the covariates, which is particularly important in high-dimensional settings where direct partition-based diagnostics can be noisy.

The ablation results in Appendix A.5 further clarify the source of these gains and the finite-sample stability of CPCP. The RCP-MultiHead baseline, which only co-trains the three quantile heads without using them to form density weights or to fine-tune the target quantile, performs comparably to RCP; hence the improvement is not merely due to multitask quantile training or additional model capacity. In contrast, CPCP improves upon RCP, indicating that the density-weighted objective is the key component. The stabilized variant CPCP (Clip+Mix) further improves most conditional-coverage metrics by controlling the variance induced by extreme reciprocal weights while retaining the benefit of reweighting. The results with $\delta = 0.01$ and $\delta = 0.05$ are also comparable to the default $\delta = 0.02$, suggesting that the method is robust to moderate changes in the finite-difference bandwidth. Finally, Table 11 reports the mean per-dimension log volume, showing that CPCP achieves substantial improvements across conditional-coverage metrics while maintaining a moderate prediction-set size, rather than simply inflating the prediction sets.

*Table 1.* Comparison of MSCE. Mean Squared Coverage Error (MSCE, ↓) on 8 high-dimensional benchmarks. Best results are **bolded**.

| Method | Bike | Diamond | Gas Turbine | Naval | SGEMM | Superconductivity | Transcoding | WEC |
|---|---|---|---|---|---|---|---|---|
| Split | 0.0031 ± 0.0008 | 0.0118 ± 0.0035 | 0.0033 ± 0.0006 | 0.0351 ± 0.0064 | 0.0039 ± 0.0010 | 0.0082 ± 0.0010 | 0.0125 ± 0.0031 | 0.0123 ± 0.0019 |
| PLCP | 0.0021 ± 0.0011 | 0.0032 ± 0.0033 | 0.0008 ± 0.0009 | 0.0162 ± 0.0087 | 0.0026 ± 0.0013 | 0.0017 ± 0.0008 | 0.0027 ± 0.0038 | 0.0031 ± 0.0033 |
| Gaussian-Scoring | 0.0011 ± 0.0004 | 0.0006 ± 0.0003 | **0.0004 ± 0.0002** | 0.0129 ± 0.0095 | 0.0041 ± 0.0041 | 0.0026 ± 0.0011 | 0.0016 ± 0.0014 | 0.0057 ± 0.0026 |
| CQR | 0.0011 ± 0.0007 | 0.0010 ± 0.0004 | 0.0006 ± 0.0002 | 0.0120 ± 0.0064 | 0.0012 ± 0.0005 | 0.0009 ± 0.0005 | 0.0016 ± 0.0009 | 0.0061 ± 0.0007 |
| CQR-ALD | **0.0008 ± 0.0003** | 0.0007 ± 0.0007 | 0.0006 ± 0.0003 | 0.0055 ± 0.0059 | 0.0009 ± 0.0006 | 0.0015 ± 0.0010 | 0.0021 ± 0.0029 | 0.0045 ± 0.0009 |
| RCP | 0.0010 ± 0.0005 | 0.0013 ± 0.0006 | 0.0006 ± 0.0004 | 0.0029 ± 0.0020 | 0.0007 ± 0.0003 | 0.0013 ± 0.0011 | 0.0009 ± 0.0003 | 0.0030 ± 0.0005 |
| RCP-ALD | 0.0013 ± 0.0010 | 0.0010 ± 0.0006 | 0.0005 ± 0.0003 | 0.0080 ± 0.0102 | 0.0009 ± 0.0007 | 0.0011 ± 0.0005 | 0.0010 ± 0.0007 | 0.0025 ± 0.0008 |
| CPCP | 0.0009 ± 0.0004 | 0.0009 ± 0.0004 | **0.0004 ± 0.0002** | 0.0019 ± 0.0010 | 0.0003 ± 0.0002 | 0.0014 ± 0.0011 | 0.0009 ± 0.0006 | 0.0025 ± 0.0009 |
| CPCP (Clip+Mix) | **0.0008 ± 0.0002** | **0.0004 ± 0.0002** | **0.0004 ± 0.0002** | 0.0019 ± 0.0009 | 0.0003 ± 0.0001 | **0.0007 ± 0.0004** | **0.0004 ± 0.0002** | **0.0012 ± 0.0004** |

*Table 2.* Comparison of WSC. Worst Slice Coverage (WSC, ↑) on 8 high-dimensional benchmarks. Best results are **bolded**.

| Method | Bike | Diamond | Gas Turbine | Naval | SGEMM | Superconductivity | Transcoding | WEC |
|---|---|---|---|---|---|---|---|---|
| Split | 0.8133 ± 0.0277 | 0.6480 ± 0.0206 | 0.8192 ± 0.0159 | 0.5428 ± 0.0446 | 0.7435 ± 0.0156 | 0.7712 ± 0.0266 | 0.6797 ± 0.0234 | 0.7623 ± 0.0195 |
| PLCP | 0.8559 ± 0.0341 | 0.8068 ± 0.0680 | 0.8690 ± 0.0229 | 0.6837 ± 0.0623 | 0.7891 ± 0.0514 | 0.8529 ± 0.0268 | 0.8123 ± 0.0645 | 0.8413 ± 0.0369 |
| Gaussian-Scoring | 0.8611 ± 0.0278 | 0.8613 ± 0.0170 | 0.8804 ± 0.0166 | 0.6928 ± 0.0911 | 0.7943 ± 0.0646 | 0.8434 ± 0.0232 | 0.8429 ± 0.0355 | 0.8259 ± 0.0347 |
| CQR | 0.8641 ± 0.0245 | 0.8563 ± 0.0159 | 0.8801 ± 0.0217 | 0.6997 ± 0.0837 | 0.8393 ± 0.0201 | 0.8704 ± 0.0235 | 0.8175 ± 0.0253 | 0.8149 ± 0.0188 |
| CQR-ALD | 0.8696 ± 0.0246 | 0.8735 ± 0.0259 | 0.8807 ± 0.0156 | 0.7836 ± 0.0745 | 0.8513 ± 0.0364 | 0.8622 ± 0.0246 | 0.8363 ± 0.0227 | 0.8222 ± 0.0193 |
| RCP | 0.8849 ± 0.0240 | 0.8448 ± 0.0237 | 0.8752 ± 0.0158 | 0.8002 ± 0.0457 | 0.8627 ± 0.0149 | 0.8638 ± 0.0260 | 0.8515 ± 0.0196 | 0.8516 ± 0.0198 |
| RCP-ALD | 0.8743 ± 0.0250 | 0.8558 ± 0.0178 | 0.8872 ± 0.0174 | 0.7596 ± 0.0803 | 0.8597 ± 0.0331 | **0.8779 ± 0.0181** | 0.8561 ± 0.0226 | 0.8582 ± 0.0173 |
| CPCP | 0.8820 ± 0.0132 | 0.8589 ± 0.0216 | 0.8844 ± 0.0125 | 0.8169 ± 0.0408 | 0.8864 ± 0.0119 | 0.8733 ± 0.0251 | 0.8548 ± 0.0172 | 0.8462 ± 0.0266 |
| CPCP (Clip+Mix) | **0.8882 ± 0.0250** | **0.8802 ± 0.0099** | **0.8912 ± 0.0139** | **0.8320 ± 0.0304** | **0.8912 ± 0.0126** | 0.8771 ± 0.0220 | **0.8759 ± 0.0158** | **0.8715 ± 0.0141** |

*Table 3.* Comparison of $L_1$-ERT. $L_1$ Excess Risk of the Target coverage ($L_1$-ERT, ↓) on 8 benchmarks. Best results are **bolded**.

| Method | Bike | Diamond | Gas Turbine | Naval | SGEMM | Superconductivity | Transcoding | WEC |
|---|---|---|---|---|---|---|---|---|
| Split | 0.0602 ± 0.0071 | 0.1223 ± 0.0050 | 0.0374 ± 0.0034 | 0.1536 ± 0.0092 | 0.1044 ± 0.0046 | 0.0956 ± 0.0058 | 0.1120 ± 0.0058 | 0.1079 ± 0.0031 |
| PLCP | 0.0424 ± 0.0167 | 0.0642 ± 0.0310 | 0.0198 ± 0.0075 | 0.0789 ± 0.0279 | 0.0809 ± 0.0295 | 0.0466 ± 0.0092 | 0.0573 ± 0.0262 | 0.0656 ± 0.0182 |
| Gaussian-Scoring | 0.0337 ± 0.0086 | 0.0330 ± 0.0057 | 0.0168 ± 0.0079 | 0.0826 ± 0.0327 | 0.0733 ± 0.0349 | 0.0443 ± 0.0105 | 0.0371 ± 0.0146 | 0.0996 ± 0.0089 |
| CQR | 0.0307 ± 0.0136 | 0.0384 ± 0.0082 | 0.0170 ± 0.0070 | 0.0783 ± 0.0282 | 0.0506 ± 0.0112 | 0.0270 ± 0.0099 | 0.0510 ± 0.0117 | 0.0762 ± 0.0035 |
| CQR-ALD | 0.0274 ± 0.0114 | 0.0301 ± 0.0131 | 0.0177 ± 0.0041 | 0.0476 ± 0.0268 | 0.0404 ± 0.0204 | 0.0328 ± 0.0127 | 0.0461 ± 0.0092 | 0.0776 ± 0.0051 |
| RCP | 0.0208 ± 0.0076 | 0.0446 ± 0.0083 | 0.0155 ± 0.0067 | 0.0353 ± 0.0184 | 0.0371 ± 0.0099 | 0.0329 ± 0.0108 | 0.0342 ± 0.0078 | 0.0574 ± 0.0052 |
| RCP-ALD | 0.0274 ± 0.0142 | 0.0373 ± 0.0100 | 0.0157 ± 0.0051 | 0.0557 ± 0.0343 | 0.0427 ± 0.0185 | 0.0309 ± 0.0084 | 0.0322 ± 0.0132 | 0.0495 ± 0.0059 |
| CPCP | 0.0195 ± 0.0097 | 0.0379 ± 0.0105 | 0.0129 ± 0.0047 | 0.0271 ± 0.0148 | 0.0166 ± 0.0062 | 0.0370 ± 0.0120 | 0.0295 ± 0.0076 | 0.0632 ± 0.0090 |
| CPCP (Clip+Mix) | **0.0178 ± 0.0076** | **0.0219 ± 0.0048** | **0.0116 ± 0.0067** | **0.0268 ± 0.0102** | **0.0155 ± 0.0061** | **0.0267 ± 0.0065** | **0.0182 ± 0.0050** | **0.0433 ± 0.0048** |

*Table 4.* Comparison of $L_2$-ERT. $L_2$ Excess Risk of the Target coverage ($L_2$-ERT, ↓) on 8 benchmarks. Best results are **bolded**.

| Method | Bike | Diamond | Gas Turbine | Naval | SGEMM | Superconductivity | Transcoding | WEC |
|---|---|---|---|---|---|---|---|---|
| Split | 0.0060 ± 0.0015 | 0.0287 ± 0.0029 | 0.0044 ± 0.0008 | 0.0338 ± 0.0067 | 0.0219 ± 0.0028 | 0.0108 ± 0.0018 | 0.0163 ± 0.0020 | 0.0225 ± 0.0018 |
| PLCP | 0.0030 ± 0.0023 | 0.0076 ± 0.0087 | 0.0011 ± 0.0011 | 0.0139 ± 0.0104 | 0.0139 ± 0.0087 | 0.0035 ± 0.0016 | 0.0059 ± 0.0051 | 0.0085 ± 0.0057 |
| Gaussian-Scoring | 0.0018 ± 0.0009 | 0.0024 ± 0.0009 | 0.0005 ± 0.0005 | 0.0127 ± 0.0130 | 0.0136 ± 0.0145 | 0.0024 ± 0.0012 | 0.0028 ± 0.0033 | 0.0079 ± 0.0035 |
| CQR | 0.0016 ± 0.0013 | 0.0024 ± 0.0009 | 0.0005 ± 0.0003 | 0.0104 ± 0.0069 | 0.0044 ± 0.0023 | 0.0007 ± 0.0014 | 0.0048 ± 0.0032 | 0.0084 ± 0.0011 |
| CQR-ALD | 0.0011 ± 0.0010 | 0.0016 ± 0.0017 | 0.0006 ± 0.0004 | 0.0049 ± 0.0075 | 0.0032 ± 0.0035 | 0.0013 ± 0.0016 | 0.0040 ± 0.0027 | 0.0125 ± 0.0025 |
| RCP | 0.0006 ± 0.0006 | 0.0029 ± 0.0011 | 0.0004 ± 0.0004 | 0.0023 ± 0.0020 | 0.0016 ± 0.0009 | 0.0014 ± 0.0015 | 0.0018 ± 0.0008 | 0.0048 ± 0.0011 |
| RCP-ALD | 0.0013 ± 0.0016 | 0.0022 ± 0.0011 | 0.0005 ± 0.0004 | 0.0072 ± 0.0110 | 0.0023 ± 0.0027 | 0.0012 ± 0.0010 | 0.0023 ± 0.0026 | 0.0039 ± 0.0013 |
| CPCP | 0.0006 ± 0.0006 | 0.0022 ± 0.0012 | 0.0003 ± 0.0002 | **0.0011 ± 0.0009** | 0.0004 ± 0.0003 | 0.0026 ± 0.0029 | 0.0015 ± 0.0007 | 0.0060 ± 0.0017 |
| CPCP (Clip+Mix) | **0.0004 ± 0.0005** | **0.0007 ± 0.0003** | **0.0002 ± 0.0002** | 0.0012 ± 0.0010 | **0.0003 ± 0.0003** | **0.0006 ± 0.0007** | **0.0005 ± 0.0003** | 0.0026 ± 0.0008 |

## 7. Conclusion

In this paper, we study the problem of improving conditional coverage in conformal prediction with finite samples and black-box predictors. By analyzing the mean-squared error of conditional coverage probabilities, we identify a key limitation of a ubiquitous component in conformal procedures: standard quantile regression based on the pinball loss. We show that, while the conformalization step effectively corrects coverage bias, variance reduction requires a mechanism that more precisely captures the underlying heteroscedastic structure. Motivated by this observation, we develop a Taylor-based approximation that recovers this structure within the conditional-coverage MSE. Building on this insight, we propose a principled algorithmic framework with exact non-asymptotic excess-risk guarantees. Extensive experiments demonstrate the superior performance of our method, and ablation studies further validate the contribution of each component.

We emphasize that the primary value of conformal prediction lies in the finite-sample regime, and accordingly adopt a lightweight modeling strategy by employing joint quantile regression rather than explicit density estimation on the calibration set. We hope our results provide useful insights for the design of practical conformal prediction procedures.

## Impact Statement

This paper advances the theoretical and algorithmic foundations of conformal prediction by improving conditional coverage in finite-sample settings. Reliable uncertainty quantification is an important component in many real-world decision-making systems, including scientific modeling, engineering, and data-driven policy analysis, where inaccurate uncertainty estimates may lead to overconfident or unsafe decisions. By providing methods with exact non-asymptotic guarantees and improved conditional calibration, our work

contributes to the robustness and reliability of machine learning models.

The proposed methods are general-purpose and do not target any specific application domain involving sensitive personal data or automated decision-making about individuals. As such, we do not anticipate any direct negative ethical or societal impacts arising from this work. We hope that our results will support the responsible deployment of machine learning systems by enabling more reliable uncertainty estimates in practice.

## Acknowledgement

This research was supported by the National Natural Science Foundation of China (No. 72171131 and No. 72133002).

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

# A. Experiments Details

## A.1. Dataset Description

We introduce our benchmarks in alphabetical order, following the sequence presented in our tables. First, we provide a summary of all eight datasets, after which we detail each one individually. We note that the majority of these datasets are sourced from the UCI repository (Kelly et al., 2025).

*Table 5.* Overview of Datasets. The feature dimension ($X$), label dimension ($Y$), and sample size ($N$) correspond to the preprocessed data used in the experiments.

| Dataset | Feature Dim. ($X$) | Label Dim. ($Y$) | Sample Size ($N$) |
|---|---|---|---|
| Bike Sharing[1] | 13 | 1 | 17,379 |
| Diamonds[2] | 23 | 1 | 53,940 |
| Gas Turbine[3] | 9 | 2 | 36,733 |
| Naval Propulsion[4] | 16 | 2 | 11,934 |
| SGEMM Product[5] | 14 | 4 | 50,000 |
| Superconductivity[6] | 81 | 1 | 21,263 |
| Transcoding[7] | 23 | 2 | 68,784 |
| WEC (Perth)[8] | 98 | 49 | 27,000 |

[1] https://archive.ics.uci.edu/dataset/275/bike+sharing+dataset
[2] https://www.kaggle.com/datasets/shivam2503/diamonds
[3] https://archive.ics.uci.edu/dataset/551/gas+turbine+co+and+nox+emission+data+set
[4] https://archive.ics.uci.edu/dataset/316/condition+based+maintenance+of+naval+propulsion+plants
[5] https://archive.ics.uci.edu/dataset/440/sgemm+gpu+kernel+performance
[6] https://archive.ics.uci.edu/dataset/464/superconductivty+data
[7] https://archive-beta.ics.uci.edu/dataset/335/online+video+characteristics+and+transcoding+time+dataset
[8] https://archive.ics.uci.edu/dataset/882/large-scale+wave+energy+farm

**Bike Sharing.** This dataset pertains to urban transportation and mobility. To prevent label leakage, we removed temporal identifiers and the sub-category counts (`casual` and `registered`) from the input features. The regression target is the total count of rental bikes (`cnt`) recorded in the system.

**Diamonds.** Originating from the field of gemology and economics, this dataset is used to predict the market value of diamonds. We applied one-hot encoding to the categorical features (`cut`, `color`, `clarity`), dropping the first category to avoid multicollinearity. The label is the price of the diamond in US dollars.

**Gas Turbine.** Focusing on environmental emissions in the energy sector, this dataset aggregates sensor data from a gas turbine power plant over the period 2011–2015. The input features include various sensor measurements, while the label is a multi-dimensional vector representing Carbon Monoxide (CO) and Nitrogen Oxides (NOx) emissions.

**Naval Propulsion.** Used for predictive maintenance in naval engineering, this dataset contains data generated from a gas turbine propulsion plant simulator. The task involves predicting a two-dimensional label corresponding to the decay state coefficients of the GT compressor and the GT turbine.

**SGEMM Product.** In the domain of high-performance computing, this dataset measures the performance of matrix multiplication kernels on GPUs. Due to the massive scale of the original data, we randomly sampled $N = 50,000$ instances for our experiments. The features represent kernel parameters, and the label consists of the execution times from four independent runs.

**Superconductivity.** This dataset comes from materials science and condensed matter physics. The input features are derived from the chemical formula and elemental properties (e.g., atomic mass, density) of various materials. The scalar regression target is the critical temperature ($T_c$) at which the material exhibits superconductivity.

**Transcoding.** Addressing cloud resource management for video processing, this dataset characterizes video transcoding tasks. We removed non-informative identifiers (such as `id` and `url`) and one-hot encoded the video codec attributes. The label is a dual-target variable comprising the transcoding time and memory usage.

**WEC (Perth).** Situated in the renewable energy domain, this dataset simulates the performance of a wave energy farm. The input features consist of the coordinate positions $(X, Y)$ for 49 floating buoys. The target is a high-dimensional vector ($d_y = 49$) representing the power output generated by each individual buoy.

### A.2. Baseline Details

With the exception of CQR, all methods utilize the training set to learn $\hat{\mu}$ via mean squared error (MSE) minimization. The estimator $\hat{\mu}$ is parameterized by a three-layer MLP with ReLU activation. The nonconformity score is given by:

$$S(X, Y) = \|Y - \hat{\mu}(X)\|_\infty, \tag{27}$$

reducing to the standard absolute residual $S(X, Y) = |Y - \hat{\mu}(X)|$ for univariate labels.

As standard Split Conformal Prediction and Rectified Conformal Prediction (RCP) have been detailed in Section 2, we proceed to introduce the other main baselines.

**PLCP.** Partition Learning Conformal Prediction (PLCP; Kiyani et al., 2024) aims to improve upon the marginal nature of Split CP by partitioning the feature space into **learned** regions. It optimizes these partitions to group samples of similar difficulty and applies Split CP locally within each group. This approach is formulated as a co-training procedure:

$$h^\star, \boldsymbol{q}^\star = \underset{\boldsymbol{q} \in \mathbb{R}^G, h \in \mathcal{H}}{\operatorname{argmin}} \frac{1}{n} \sum_{j=1}^{n} \sum_{i=1}^{G} h^i(X_j) \rho_\tau(q_i, S_j), \tag{28}$$

where $G$ denotes the number of distinct groups (we use $G$ to avoid conflict with the calibration sample size $m$, differing from the notation in Kiyani et al., 2024), and $h : \mathcal{X} \to \Delta_G$ is a partition function assigning samples to groups.

Since $G$ is finite, provided that $h$ converges, the resulting $\boldsymbol{q}$ corresponds exactly to the empirical $\tau$-quantile for each group. Consequently, PLCP avoids the issue discussed at the start of Section 5, namely the conformalization shift $\hat{\gamma}$ often associated with quantile regression. We report results for $G = 20$ in the main paper and provide supplementary results for $G = 50$ in Appendix A.5.

**Gaussian Scoring.** As described in (Braun et al., 2026), this method assumes a multivariate Gaussian density for $Y \mid X$. On the training set, we train two MLPs to simultaneously predict the conditional mean $\hat{\mu}(X)$ and covariance matrix $\hat{\Sigma}(X)$ by minimizing the negative log-likelihood (NLL). The nonconformity score is defined as the Mahalanobis distance:

$$S_{\mathrm{Mah}}(X, Y) = \left\| \hat{\Sigma}(X)^{-1/2} (Y - \hat{\mu}(X)) \right\|_2, \tag{29}$$

which reduces to the standard normalized residual $S(X, Y) = |Y - \hat{\mu}(X)|/\hat{\sigma}(X)$ (Lei et al., 2018) when $Y$ is one-dimensional.

**CQR.** Conformalized Quantile Regression (CQR; Romano et al., 2019) constructs adaptive intervals by directly estimating conditional quantiles on the training set, rather than estimating the conditional mean $\hat{\mu}$. We train two MLPs to output the lower ($\hat{q}_{\alpha/2}$) and upper ($\hat{q}_{1-\alpha/2}$) quantiles by minimizing the pinball loss. The nonconformity score is defined as:

$$S(X, Y) = \max(\hat{q}_{\alpha/2}(X) - Y, Y - \hat{q}_{1-\alpha/2}(X)). \tag{30}$$

We extend CQR to multivariate labels $Y \in \mathbb{R}^d$ by predicting the quantiles for each dimension simultaneously. In this setting, the nonconformity score is the maximum signed distance to the predicted interval boundaries across all dimensions:

$$S(X, Y) = \max_{j \in [d]} \max(\hat{q}^{(j)}_{\alpha/2}(X) - Y^{(j)}, Y^{(j)} - \hat{q}^{(j)}_{1-\alpha/2}(X)). \tag{31}$$

**CQR/RCP-ALD.** We begin by introducing the Asymmetric Laplace Distribution (ALD). Given a location parameter $\mu$, scale parameter $\sigma$, and skewness parameter $\tau$, the PDF of the ALD is:

$$f(y \mid \mu, \sigma, \tau) = \frac{\tau(1-\tau)}{\sigma} \exp\left(-\frac{\rho_\tau(\mu, y)}{\sigma}\right). \tag{32}$$

Adapting this to model the conditional distribution $S \mid X$, we express the maximum likelihood estimation (MLE) in our notation as:

$$\max_{\hat{\sigma}(x), \hat{q}(x)} \frac{1}{n} \sum_{i=1}^{n} \left(-\ln(\hat{\sigma}(x_i)) - \frac{\rho_\tau(\hat{q}(x_i), s_i)}{\hat{\sigma}(x_i)}\right). \tag{33}$$

As discussed in Example 3.1, we select the ALD as a representative of the location-scale family. Notably, the MLE for this parametric model corresponds to a weighted pinball loss, where the weights are the inverse scale estimates $1/\hat{\sigma}(x)$. We incorporate this into CQR and RCP by replacing the standard quantile regression (pinball loss) with the ALD-based likelihood, yielding two additional baselines denoted by the suffix *-ALD*. Although we do not assume the data strictly follow an ALD, we employ this formulation as a quasi-likelihood to derive the objective function.

These two baselines are designed to isolate and examine the benefits of the approximation constructed in Proposition 3.4, effectively serving as an ablation study.

**RCP-Multihead.** We use this baseline to examine the impact of co-training three quantiles, i.e., the multitasking step. In this baseline, we only co-train the three quantiles and use the central quantile estimator, $\hat{q}_\tau$, to implement the subsequent RCP procedures.

## A.3. CPCP Details

In this section, we provide a detailed description of the implementation, network architecture, and optimization procedure for our proposed method, **Colorful Pinball Conformal Prediction (CPCP)**.

**Network architecture.** To ensure the structural constraint that quantile estimates remain non-crossing (i.e., $\hat{q}_{\text{low}}(x) \leq \hat{q}_{\text{main}}(x) \leq \hat{q}_{\text{high}}(x)$), we implement a specialized neural network architecture named `MonotonicThreeHeadNet`.

The network consists of a shared feature extractor and three specific heads:

- **Shared feature extractor:** An MLP with two hidden layers (256 units each) and ReLU activation functions.

- **Main head:** A linear layer that outputs the central quantile estimate $\hat{q}_{\text{main}}(x)$.

- **Auxiliary heads:** Two separate linear layers followed by a `Softplus` activation function to predict the positive gaps $\Delta_{\text{low}}(x)$ and $\Delta_{\text{high}}(x)$.

The final outputs are constructed as:

$$\hat{q}_{\text{low}}(x) = \hat{q}_{\text{main}}(x) - \text{Softplus}(\Delta_{\text{low}}(x)), \tag{34}$$
$$\hat{q}_{\text{high}}(x) = \hat{q}_{\text{main}}(x) + \text{Softplus}(\Delta_{\text{high}}(x)). \tag{35}$$

This parameterization strictly guarantees monotonicity and ensures that the denominator in our weight calculation remains positive.

**Data splitting.** Following Algorithm 1, we partition the calibration dataset $\mathcal{D}_{\text{cal}}$ into three disjoint subsets:

- $\mathcal{D}_{\text{cal},1}$ (Base Training): 40% of the calibration data, used to train the three initial quantile heads.

- $\mathcal{D}_{\text{cal},2}$ (Fine-tuning): 40% of the calibration data, used to fine-tune the main head with estimated density weights.

- $\mathcal{D}_{\text{cal},3}$ (Conformalization): 20% of the calibration data, used to compute the final rectified conformity scores $R_j$ and derive the predictive interval.

**Clipping and Mixing.** To prevent extreme weights from destabilizing training, we implement two regularization techniques:

- **Clipping:** The **normalized** weights are clamped to a maximum value (default $M = 5.0$), as detailed in Algorithm 2.

- **Loss Mixing:** The final objective function is a convex combination of the weighted loss and the original pinball loss:

$$\mathcal{L}_{\text{total}} = \lambda \mathcal{L}_{\text{weighted}} + (1 - \lambda)\mathcal{L}_{\text{pinball}}, \tag{36}$$

where $\lambda$ is the mixing ratio (default $\lambda = 0.5$).

Effectively, these two strategies impose an artificial lower bound (via $\lambda = 0.5$) and upper bound ($M = 5.0$) on the normalized weights.

**Quantile gap.** Although we derive the non-asymptotic expression $\delta(n)$, we take $\delta = 0.02$ as default throughout our experiments. For comparison, we also provide the results with $\delta = 0.01, 0.05$, which we denote by CPCP-0.01 and CPCP-0.05, respectively.

In general, a smaller $\delta$ reduces the bias of the finite-difference estimation but requires a larger sample size to maintain stability. Our theory establishes that the optimal rate for this trade-off is $\delta^{\star} \asymp n^{-1/6}$.

---

**Algorithm 2** Colorful Pinball Conformal Prediction (Clipped Version)

---

1: **Input:** Calibration data $\mathcal{D}_{\text{cal}}$, Point predictor $\hat{\mu}$, Test input $x_{\text{test}}$, Target coverage level $\tau = 1 - \alpha$, Bandwidth $\delta$, Clipping multiplier $M$
2: Split $\mathcal{D}_{\text{cal}}$ into three disjoint parts $\mathcal{D}_{\text{cal},1}$, $\mathcal{D}_{\text{cal},2}$, $\mathcal{D}_{\text{cal},3}$.
3: Compute conformity scores $S_i$ for all $(x_i, y_i) \in \mathcal{D}_{\text{cal}}$.
4: Train three quantile estimators $\hat{q}_{\text{low}}$, $\hat{q}_{\text{main}}$, and $\hat{q}_{\text{high}}$ (for levels $\tau - \delta, \tau, \tau + \delta$) jointly on $\mathcal{D}_{\text{cal},1}$.
5: Compute the raw weight for all $(x_i, y_i) \in \mathcal{D}_{\text{cal},2}$:

$$w_i \leftarrow \frac{2\delta}{\hat{q}_{\text{high}}(x_i) - \hat{q}_{\text{low}}(x_i)}.$$

6: Clip and normalize the weights using threshold $\tau \leftarrow M \cdot \text{mean}(\{w_j\}_{j \in \mathcal{D}_{\text{cal},2}})$:

$$w_i \leftarrow \min(w_i, \tau), \quad w_i \leftarrow w_i / \sum_{j \in \mathcal{D}_{\text{cal},2}} w_j.$$

7: Fine-tune the parameters of $\hat{q}_{\text{main}}$ on $\mathcal{D}_{\text{cal},2}$ using weights $w_i$ to get the final quantile estimator $\hat{q}_{\tau}$.
8: Compute residuals for all $j \in \mathcal{D}_{\text{cal},3}$:
$$R_j \leftarrow S_j - \hat{q}_{\tau}(x_j)$$
9: Compute empirical quantile $\hat{\gamma}$ as the $\lceil(|\mathcal{D}_{\text{cal},3}| + 1)(1 - \alpha)\rceil$-th smallest value of $\{R_j\}$.
10: **Output:** $\mathcal{C}_{\alpha}(x_{\text{test}}) = \{y : S(x_{\text{test}}, y) \leq \hat{\gamma} + \hat{q}_{\tau}(x_{\text{test}})\}$

---

### A.4. Metric Details

In summary, we consider two approaches to approximate coverage performance, based on clustering and random projections, respectively.

**MSCE**   For MSCE, we use K-means to partition samples into $K$ groups. There is a fundamental trade-off here: increasing $K$ approximates the conditional coverage with finer granularity, but reduces the sample size within each group, thereby decreasing the precision of the empirical coverage estimate. Our guiding principle is to ensure sufficient samples (on the order of hundreds) within each group to evaluate coverage with percent-level precision. Consequently, we examine two settings: $K = 10$ and $K = 30$, as reported in Tables 6 and 7.

**WSC**   Romano et al. (2020); Cauchois et al. (2021) propose worst-slab coverage (WSC). We first define a slab in the covariate space as:

$$S_{v,a,b} = \left\{ x \in \mathbb{R}^d \mid a \le v^T x \le b \right\},$$

where the direction $v \in \mathbb{R}^d$ and scalars $a < b$ are chosen to define the slice. For a predictive set $\mathcal{C}_\alpha$ and a threshold $\eta \in (0, 1)$, the WSC is defined as the minimum coverage over all sufficiently large slabs:

$$\text{WSC}(\mathcal{C}_\alpha; \eta) = \inf_{v \in \mathbb{R}^d, a < b} \left\{ \mathbb{P}\left[ Y \in \mathcal{C}_\alpha(X) \mid X \in S_{v,a,b} \right] \mid \mathbb{P}\left[ X \in S_{v,a,b} \right] \ge 1 - \eta \right\}. \tag{37}$$

In practice, we estimate WSC for a given $\mathcal{C}_\alpha$ by sampling 1,000 independent vectors $v$ from the unit sphere in $\mathbb{R}^d$ and optimizing parameters $a$ and $b$ via grid search. To mitigate finite-sample negative bias, we partition the test data into two disjoint subsets (e.g., with a 25%-75% split). We use the first subset to determine the optimal parameters $v^\star, a^\star, b^\star$, and the second to evaluate the conditional coverage:

$$\mathbb{P}\left[ Y \in \mathcal{C}_\alpha(X) \mid X \in S_{v^\star, a^\star, b^\star} \right].$$

**$\ell$-ERT**   Braun et al. (2025) introduce the Excess Risk of the Target coverage (ERT), which casts coverage evaluation as a binary classification problem. This approach relies on the insight that under perfect conditional coverage, the coverage indicator $Z = \mathbb{I}\{Y \in \mathcal{C}_\alpha(X)\}$ behaves as a Bernoulli variable with constant probability $1 - \alpha$, independent of $X$. Consequently, any classifier $h(X)$ that achieves a lower risk than the constant predictor $1 - \alpha$ exposes a violation of conditional validity. The metric is defined as the risk reduction:

$$\ell\text{-ERT} = \mathcal{R}_\ell(1 - \alpha) - \mathcal{R}_\ell(h), \tag{38}$$

where $\mathcal{R}_\ell$ is the risk under a proper loss function $\ell$. Depending on the choice of $\ell$, this metric estimates different distances; specifically, we consider $L_1$-ERT and $L_2$-ERT, which provide lower-bound estimates for $\mathbb{E}[|\pi(X) - \tau|]$ and $\mathbb{E}[(\pi(X) - \tau)^2]$ (i.e., MSCE), respectively. In our experiments, we employ logistic regression as the classifier $h$ to estimate these quantities.

**Volume.**   We report the average log-volume per dimension, calculated as $\frac{1}{d} \log \text{Volume}$. Because we use the infinity norm for the nonconformity score, the predictive sets for all baselines (except Gaussian scoring) are hyper-rectangles. Gaussian scoring, in contrast, produces hyper-ellipsoidal sets. Since ellipsoids are inherently more compact than hyper-rectangles, Gaussian scoring enjoys a natural advantage on this metric.

### A.5. Detailed Results

*Table 6.* Full comparison of MSCE ($K = 10$) . MSCE ($\downarrow$) on all benchmarks with all baselines.

| Method | Bike | Diamond | Gas Turbine | Naval | SGEMM | Superconductivity | Transcoding | WEC |
|---|---|---|---|---|---|---|---|---|
| Split | 0.0031 ± 0.0008 | 0.0118 ± 0.0035 | 0.0033 ± 0.0006 | 0.0351 ± 0.0064 | 0.0039 ± 0.0010 | 0.0082 ± 0.0010 | 0.0125 ± 0.0031 | 0.0123 ± 0.0019 |
| PLCP (G=20) | 0.0021 ± 0.0011 | 0.0032 ± 0.0033 | 0.0008 ± 0.0009 | 0.0162 ± 0.0087 | 0.0026 ± 0.0013 | 0.0017 ± 0.0008 | 0.0027 ± 0.0038 | 0.0031 ± 0.0033 |
| PLCP (G=50) | 0.0016 ± 0.0007 | 0.0016 ± 0.0022 | 0.0008 ± 0.0010 | 0.0124 ± 0.0075 | 0.0015 ± 0.0015 | 0.0015 ± 0.0009 | 0.0009 ± 0.0006 | 0.0017 ± 0.0015 |
| Gaussian-Scoring | 0.0011 ± 0.0004 | 0.0006 ± 0.0003 | **0.0004 ± 0.0002** | 0.0129 ± 0.0095 | 0.0041 ± 0.0041 | 0.0026 ± 0.0011 | 0.0016 ± 0.0014 | 0.0057 ± 0.0026 |
| CQR | 0.0011 ± 0.0007 | 0.0010 ± 0.0004 | 0.0006 ± 0.0002 | 0.0120 ± 0.0064 | 0.0012 ± 0.0005 | 0.0009 ± 0.0005 | 0.0016 ± 0.0009 | 0.0061 ± 0.0007 |
| CQR-ALD | 0.0008 ± 0.0003 | 0.0007 ± 0.0007 | 0.0006 ± 0.0003 | 0.0055 ± 0.0059 | 0.0009 ± 0.0006 | 0.0015 ± 0.0010 | 0.0021 ± 0.0029 | 0.0045 ± 0.0009 |
| RCP | 0.0010 ± 0.0005 | 0.0013 ± 0.0006 | 0.0006 ± 0.0004 | 0.0029 ± 0.0020 | 0.0007 ± 0.0003 | 0.0013 ± 0.0011 | 0.0009 ± 0.0003 | 0.0030 ± 0.0005 |
| RCP-ALD | 0.0013 ± 0.0010 | 0.0010 ± 0.0006 | 0.0005 ± 0.0003 | 0.0080 ± 0.0102 | 0.0009 ± 0.0007 | 0.0011 ± 0.0005 | 0.0010 ± 0.0007 | 0.0025 ± 0.0008 |
| RCP-MultiHead | 0.0010 ± 0.0004 | 0.0015 ± 0.0006 | 0.0005 ± 0.0003 | 0.0027 ± 0.0013 | 0.0010 ± 0.0004 | 0.0011 ± 0.0008 | 0.0008 ± 0.0005 | 0.0031 ± 0.0008 |
| CPCP | 0.0009 ± 0.0004 | 0.0009 ± 0.0004 | **0.0004 ± 0.0002** | 0.0019 ± 0.0010 | 0.0003 ± 0.0002 | 0.0014 ± 0.0011 | 0.0009 ± 0.0006 | 0.0025 ± 0.0009 |
| CPCP (Clip) | 0.0010 ± 0.0004 | 0.0005 ± 0.0003 | 0.0005 ± 0.0003 | 0.0020 ± 0.0013 | 0.0003 ± 0.0002 | 0.0011 ± 0.0006 | 0.0007 ± 0.0004 | 0.0016 ± 0.0006 |
| CPCP (Mix) | 0.0009 ± 0.0005 | 0.0006 ± 0.0004 | **0.0004 ± 0.0002** | 0.0014 ± 0.0007 | 0.0004 ± 0.0001 | 0.0009 ± 0.0005 | 0.0006 ± 0.0003 | 0.0016 ± 0.0005 |
| CPCP (Clip+Mix) | 0.0008 ± 0.0002 | **0.0004 ± 0.0002** | **0.0004 ± 0.0002** | 0.0019 ± 0.0009 | 0.0003 ± 0.0001 | 0.0007 ± 0.0004 | **0.0004 ± 0.0002** | 0.0012 ± 0.0004 |
| CPCP-0.01 | 0.0010 ± 0.0004 | 0.0007 ± 0.0004 | 0.0005 ± 0.0002 | 0.0023 ± 0.0012 | **0.0002 ± 0.0001** | 0.0009 ± 0.0005 | 0.0010 ± 0.0006 | 0.0024 ± 0.0008 |
| CPCP-0.01 (Clip+Mix) | 0.0008 ± 0.0004 | **0.0004 ± 0.0003** | **0.0004 ± 0.0002** | 0.0012 ± 0.0007 | 0.0003 ± 0.0001 | **0.0006 ± 0.0004** | **0.0004 ± 0.0002** | 0.0013 ± 0.0005 |
| CPCP-0.05 | 0.0008 ± 0.0004 | 0.0008 ± 0.0005 | **0.0004 ± 0.0002** | 0.0017 ± 0.0009 | 0.0003 ± 0.0002 | 0.0009 ± 0.0004 | 0.0010 ± 0.0007 | 0.0026 ± 0.0013 |
| CPCP-0.05 (Clip+Mix) | **0.0007 ± 0.0003** | 0.0005 ± 0.0002 | **0.0004 ± 0.0002** | 0.0016 ± 0.0010 | 0.0003 ± 0.0002 | **0.0006 ± 0.0003** | **0.0004 ± 0.0003** | **0.0008 ± 0.0003** |

*Table 7.* Full comparison of MSCE ($K = 30$). MSCE ($\downarrow$) on all benchmarks with all baselines.

| Method | Bike | Diamond | Gas Turbine | Naval | SGEMM | Superconductivity | Transcoding | WEC |
|---|---|---|---|---|---|---|---|---|
| Split | 0.0053 ± 0.0013 | 0.0138 ± 0.0021 | 0.0046 ± 0.0008 | 0.0392 ± 0.0070 | 0.0058 ± 0.0010 | 0.0089 ± 0.0012 | 0.0147 ± 0.0025 | 0.0186 ± 0.0016 |
| PLCP (G=20) | 0.0037 ± 0.0015 | 0.0045 ± 0.0041 | 0.0017 ± 0.0010 | 0.0200 ± 0.0078 | 0.0038 ± 0.0018 | 0.0040 ± 0.0012 | 0.0043 ± 0.0041 | 0.0062 ± 0.0055 |
| PLCP (G=50) | 0.0032 ± 0.0009 | 0.0025 ± 0.0036 | 0.0017 ± 0.0012 | 0.0155 ± 0.0082 | 0.0026 ± 0.0023 | 0.0040 ± 0.0016 | 0.0023 ± 0.0015 | 0.0036 ± 0.0024 |
| Gaussian-Scoring | 0.0023 ± 0.0006 | 0.0014 ± 0.0004 | 0.0010 ± 0.0003 | 0.0181 ± 0.0119 | 0.0056 ± 0.0052 | 0.0040 ± 0.0013 | 0.0026 ± 0.0020 | 0.0082 ± 0.0033 |
| CQR | 0.0025 ± 0.0012 | 0.0016 ± 0.0004 | 0.0011 ± 0.0003 | 0.0159 ± 0.0083 | 0.0018 ± 0.0007 | 0.0018 ± 0.0008 | 0.0034 ± 0.0014 | 0.0084 ± 0.0009 |
| CQR-ALD | 0.0019 ± 0.0005 | 0.0012 ± 0.0008 | 0.0011 ± 0.0003 | 0.0085 ± 0.0082 | 0.0014 ± 0.0009 | 0.0025 ± 0.0012 | 0.0036 ± 0.0030 | 0.0076 ± 0.0010 |
| RCP | 0.0019 ± 0.0006 | 0.0021 ± 0.0006 | 0.0012 ± 0.0004 | 0.0061 ± 0.0026 | 0.0011 ± 0.0004 | 0.0023 ± 0.0012 | 0.0016 ± 0.0003 | 0.0046 ± 0.0007 |
| RCP-ALD | 0.0023 ± 0.0010 | 0.0017 ± 0.0006 | 0.0012 ± 0.0004 | 0.0114 ± 0.0106 | 0.0014 ± 0.0009 | 0.0021 ± 0.0006 | 0.0023 ± 0.0024 | 0.0038 ± 0.0010 |
| RCP-MultiHead | 0.0020 ± 0.0004 | 0.0022 ± 0.0006 | 0.0011 ± 0.0004 | 0.0063 ± 0.0021 | 0.0015 ± 0.0004 | 0.0021 ± 0.0007 | 0.0016 ± 0.0007 | 0.0046 ± 0.0009 |
| CPCP | 0.0021 ± 0.0008 | 0.0020 ± 0.0008 | 0.0010 ± 0.0003 | 0.0054 ± 0.0019 | 0.0007 ± 0.0002 | 0.0026 ± 0.0018 | 0.0019 ± 0.0009 | 0.0043 ± 0.0013 |
| CPCP (Clip) | 0.0022 ± 0.0008 | 0.0012 ± 0.0005 | 0.0010 ± 0.0005 | 0.0056 ± 0.0019 | 0.0007 ± 0.0002 | 0.0022 ± 0.0008 | 0.0018 ± 0.0014 | 0.0035 ± 0.0007 |
| CPCP (Mix) | 0.0019 ± 0.0007 | 0.0015 ± 0.0008 | **0.0009 ± 0.0003** | 0.0046 ± 0.0014 | 0.0007 ± 0.0001 | 0.0021 ± 0.0013 | 0.0013 ± 0.0005 | 0.0029 ± 0.0008 |
| CPCP (Clip+Mix) | **0.0018 ± 0.0005** | **0.0010 ± 0.0002** | 0.0011 ± 0.0003 | 0.0054 ± 0.0019 | 0.0007 ± 0.0002 | 0.0016 ± 0.0005 | **0.0010 ± 0.0002** | 0.0026 ± 0.0008 |
| CPCP-0.01 | 0.0023 ± 0.0007 | 0.0020 ± 0.0008 | 0.0012 ± 0.0003 | 0.0062 ± 0.0018 | **0.0006 ± 0.0002** | 0.0023 ± 0.0009 | 0.0025 ± 0.0020 | 0.0042 ± 0.0011 |
| CPCP-0.01 (Clip+Mix) | **0.0018 ± 0.0005** | 0.0011 ± 0.0004 | **0.0009 ± 0.0002** | 0.0043 ± 0.0014 | 0.0007 ± 0.0002 | 0.0017 ± 0.0007 | **0.0010 ± 0.0004** | 0.0026 ± 0.0006 |
| CPCP-0.05 | **0.0018 ± 0.0004** | 0.0021 ± 0.0012 | 0.0011 ± 0.0003 | 0.0058 ± 0.0022 | 0.0007 ± 0.0002 | 0.0020 ± 0.0009 | 0.0023 ± 0.0012 | 0.0045 ± 0.0018 |
| CPCP-0.05 (Clip+Mix) | **0.0018 ± 0.0004** | **0.0010 ± 0.0002** | 0.0010 ± 0.0003 | 0.0049 ± 0.0016 | 0.0007 ± 0.0002 | **0.0015 ± 0.0003** | **0.0010 ± 0.0003** | **0.0020 ± 0.0004** |

*Table 8.* Full comparison of WSC. WSC ($\uparrow$) on all benchmarks with all baselines.

| Method | Bike | Diamond | Gas Turbine | Naval | SGEMM | Superconductivity | Transcoding | WEC |
|---|---|---|---|---|---|---|---|---|
| Split | 0.8133 ± 0.0277 | 0.6480 ± 0.0206 | 0.8192 ± 0.0159 | 0.5428 ± 0.0446 | 0.7435 ± 0.0156 | 0.7712 ± 0.0266 | 0.6797 ± 0.0234 | 0.7623 ± 0.0195 |
| PLCP (G=20) | 0.8559 ± 0.0341 | 0.8068 ± 0.0680 | 0.8690 ± 0.0229 | 0.6837 ± 0.0623 | 0.7891 ± 0.0514 | 0.8529 ± 0.0268 | 0.8123 ± 0.0645 | 0.8413 ± 0.0369 |
| PLCP (G=50) | 0.8663 ± 0.0203 | 0.8498 ± 0.0555 | 0.8738 ± 0.0162 | 0.7127 ± 0.0828 | 0.8303 ± 0.0608 | 0.8585 ± 0.0243 | 0.8450 ± 0.0208 | 0.8604 ± 0.0199 |
| Gaussian-Scoring | 0.8611 ± 0.0278 | 0.8613 ± 0.0170 | 0.8804 ± 0.0166 | 0.6928 ± 0.0911 | 0.7943 ± 0.0646 | 0.8434 ± 0.0232 | 0.8429 ± 0.0355 | 0.8259 ± 0.0347 |
| CQR | 0.8641 ± 0.0245 | 0.8563 ± 0.0159 | 0.8801 ± 0.0217 | 0.6997 ± 0.0837 | 0.8393 ± 0.0201 | 0.8704 ± 0.0235 | 0.8175 ± 0.0253 | 0.8149 ± 0.0188 |
| CQR-ALD | 0.8696 ± 0.0246 | 0.8735 ± 0.0259 | 0.8807 ± 0.0156 | 0.7836 ± 0.0745 | 0.8513 ± 0.0364 | 0.8622 ± 0.0246 | 0.8363 ± 0.0227 | 0.8222 ± 0.0193 |
| RCP | 0.8849 ± 0.0240 | 0.8448 ± 0.0237 | 0.8752 ± 0.0158 | 0.8002 ± 0.0457 | 0.8627 ± 0.0149 | 0.8638 ± 0.0260 | 0.8515 ± 0.0196 | 0.8516 ± 0.0198 |
| RCP-ALD | 0.8743 ± 0.0250 | 0.8558 ± 0.0178 | 0.8872 ± 0.0174 | 0.7596 ± 0.0803 | 0.8597 ± 0.0331 | 0.8779 ± 0.0181 | 0.8561 ± 0.0226 | 0.8582 ± 0.0173 |
| RCP-MultiHead | 0.8911 ± 0.0191 | 0.8361 ± 0.0172 | 0.8732 ± 0.0164 | 0.7985 ± 0.0286 | 0.8484 ± 0.0162 | 0.8745 ± 0.0247 | 0.8529 ± 0.0226 | 0.8519 ± 0.0150 |
| CPCP | 0.8820 ± 0.0132 | 0.8589 ± 0.0216 | 0.8844 ± 0.0125 | 0.8169 ± 0.0408 | 0.8864 ± 0.0119 | 0.8733 ± 0.0251 | 0.8548 ± 0.0172 | 0.8462 ± 0.0266 |
| CPCP (Clip) | 0.8895 ± 0.0213 | 0.8682 ± 0.0173 | 0.8805 ± 0.0175 | 0.8205 ± 0.0357 | 0.8860 ± 0.0188 | 0.8701 ± 0.0176 | 0.8706 ± 0.0191 | 0.8604 ± 0.0219 |
| CPCP (Mix) | 0.8812 ± 0.0182 | 0.8676 ± 0.0119 | 0.8816 ± 0.0141 | 0.8335 ± 0.0333 | 0.8873 ± 0.0139 | 0.8767 ± 0.0378 | 0.8736 ± 0.0137 | 0.8648 ± 0.0150 |
| CPCP (Clip+Mix) | 0.8882 ± 0.0250 | **0.8802 ± 0.0099** | **0.8912 ± 0.0139** | 0.8320 ± 0.0304 | **0.8912 ± 0.0126** | 0.8771 ± 0.0220 | 0.8759 ± 0.0158 | 0.8715 ± 0.0141 |
| CPCP-0.01 | 0.8830 ± 0.0141 | 0.8571 ± 0.0165 | 0.8794 ± 0.0126 | 0.8048 ± 0.0460 | 0.8852 ± 0.0094 | 0.8722 ± 0.0256 | 0.8581 ± 0.0229 | 0.8560 ± 0.0184 |
| CPCP-0.01 (Clip+Mix) | **0.8918 ± 0.0158** | 0.8783 ± 0.0200 | 0.8894 ± 0.0149 | **0.8390 ± 0.0357** | 0.8853 ± 0.0083 | 0.8801 ± 0.0199 | **0.8811 ± 0.0135** | 0.8700 ± 0.0211 |
| CPCP-0.05 | 0.8888 ± 0.0150 | 0.8617 ± 0.0202 | 0.8831 ± 0.0154 | 0.8252 ± 0.0399 | 0.8873 ± 0.0128 | 0.8688 ± 0.0227 | 0.8519 ± 0.0216 | 0.8435 ± 0.0254 |
| CPCP-0.05 (Clip+Mix) | 0.8915 ± 0.0162 | 0.8801 ± 0.0178 | 0.8841 ± 0.0123 | 0.8258 ± 0.0368 | 0.8865 ± 0.0129 | **0.8846 ± 0.0144** | 0.8809 ± 0.0145 | **0.8750 ± 0.0172** |

*Table 9.* Full comparison of $L_1$-ERT. $L_1$-ERT ($\downarrow$) on all benchmarks with all baselines.

| Method | Bike | Diamond | Gas Turbine | Naval | SGEMM | Superconductivity | Transcoding | WEC |
|---|---|---|---|---|---|---|---|---|
| Split | 0.0602 ± 0.0071 | 0.1223 ± 0.0050 | 0.0374 ± 0.0034 | 0.1536 ± 0.0092 | 0.1044 ± 0.0046 | 0.0956 ± 0.0058 | 0.1120 ± 0.0058 | 0.1079 ± 0.0031 |
| PLCP (G=20) | 0.0424 ± 0.0167 | 0.0642 ± 0.0310 | 0.0198 ± 0.0096 | 0.0789 ± 0.0279 | 0.0809 ± 0.0295 | 0.0466 ± 0.0092 | 0.0573 ± 0.0262 | 0.0656 ± 0.0182 |
| PLCP (G=50) | 0.0351 ± 0.0142 | 0.0445 ± 0.0241 | 0.0219 ± 0.0075 | 0.0685 ± 0.0296 | 0.0557 ± 0.0331 | 0.0455 ± 0.0100 | 0.0417 ± 0.0102 | 0.0565 ± 0.0109 |
| Gaussian-Scoring | 0.0337 ± 0.0086 | 0.0330 ± 0.0057 | 0.0168 ± 0.0079 | 0.0826 ± 0.0327 | 0.0733 ± 0.0349 | 0.0443 ± 0.0105 | 0.0371 ± 0.0146 | 0.0691 ± 0.0120 |
| CQR | 0.0307 ± 0.0136 | 0.0384 ± 0.0082 | 0.0170 ± 0.0070 | 0.0783 ± 0.0282 | 0.0506 ± 0.0112 | 0.0270 ± 0.0099 | 0.0510 ± 0.0117 | 0.0762 ± 0.0035 |
| CQR-ALD | 0.0274 ± 0.0114 | 0.0301 ± 0.0131 | 0.0177 ± 0.0041 | 0.0476 ± 0.0268 | 0.0404 ± 0.0204 | 0.0328 ± 0.0127 | 0.0461 ± 0.0092 | 0.0776 ± 0.0051 |
| RCP | 0.0208 ± 0.0076 | 0.0446 ± 0.0083 | 0.0155 ± 0.0067 | 0.0353 ± 0.0184 | 0.0371 ± 0.0099 | 0.0329 ± 0.0108 | 0.0342 ± 0.0078 | 0.0574 ± 0.0052 |
| RCP-ALD | 0.0274 ± 0.0142 | 0.0373 ± 0.0100 | 0.0157 ± 0.0051 | 0.0557 ± 0.0343 | 0.0427 ± 0.0185 | 0.0309 ± 0.0084 | 0.0322 ± 0.0132 | 0.0495 ± 0.0059 |
| RCP-MultiHead | 0.0239 ± 0.0072 | 0.0484 ± 0.0082 | 0.0179 ± 0.0077 | 0.0366 ± 0.0138 | 0.0477 ± 0.0064 | 0.0323 ± 0.0084 | 0.0335 ± 0.0112 | 0.0573 ± 0.0070 |
| CPCP | 0.0195 ± 0.0097 | 0.0379 ± 0.0105 | 0.0129 ± 0.0047 | 0.0271 ± 0.0148 | 0.0166 ± 0.0062 | 0.0370 ± 0.0120 | 0.0295 ± 0.0076 | 0.0632 ± 0.0090 |
| CPCP (Clip) | 0.0228 ± 0.0103 | 0.0256 ± 0.0065 | 0.0148 ± 0.0066 | 0.0278 ± 0.0100 | 0.0189 ± 0.0065 | 0.0346 ± 0.0059 | 0.0242 ± 0.0070 | 0.0530 ± 0.0063 |
| CPCP (Mix) | 0.0196 ± 0.0083 | 0.0275 ± 0.0094 | 0.0135 ± 0.0048 | **0.0182 ± 0.0107** | 0.0158 ± 0.0054 | 0.0319 ± 0.0092 | 0.0219 ± 0.0070 | 0.0498 ± 0.0090 |
| CPCP (Clip+Mix) | 0.0178 ± 0.0076 | **0.0219 ± 0.0048** | 0.0116 ± 0.0067 | 0.0268 ± 0.0102 | 0.0155 ± 0.0061 | 0.0267 ± 0.0065 | 0.0182 ± 0.0050 | 0.0433 ± 0.0048 |
| CPCP-0.01 | 0.0225 ± 0.0082 | 0.0394 ± 0.0108 | 0.0170 ± 0.0070 | 0.0280 ± 0.0139 | 0.0153 ± 0.0057 | 0.0362 ± 0.0113 | 0.0308 ± 0.0092 | 0.0568 ± 0.0082 |
| CPCP-0.01 (Clip+Mix) | **0.0165 ± 0.0078** | 0.0230 ± 0.0081 | **0.0105 ± 0.0051** | 0.0202 ± 0.0113 | **0.0150 ± 0.0073** | 0.0257 ± 0.0065 | 0.0177 ± 0.0045 | 0.0464 ± 0.0065 |
| CPCP-0.05 | 0.0217 ± 0.0081 | 0.0375 ± 0.0129 | 0.0140 ± 0.0070 | 0.0250 ± 0.0094 | 0.0165 ± 0.0064 | 0.0337 ± 0.0081 | 0.0312 ± 0.0079 | 0.0575 ± 0.0125 |
| CPCP-0.05 (Clip+Mix) | 0.0169 ± 0.0082 | 0.0243 ± 0.0049 | 0.0110 ± 0.0053 | 0.0221 ± 0.0128 | 0.0152 ± 0.0051 | 0.0273 ± 0.0061 | **0.0175 ± 0.0056** | **0.0376 ± 0.0049** |

*Table 10.* Full comparison of $L_2$-ERT. $L_2$-ERT ($\downarrow$) on all benchmarks with all baselines.

| Method | Bike | Diamond | Gas Turbine | Naval | SGEMM | Superconductivity | Transcoding | WEC |
|---|---|---|---|---|---|---|---|---|
| Split | 0.0060 ± 0.0015 | 0.0287 ± 0.0029 | 0.0044 ± 0.0008 | 0.0338 ± 0.0067 | 0.0219 ± 0.0028 | 0.0108 ± 0.0018 | 0.0163 ± 0.0020 | 0.0225 ± 0.0018 |
| PLCP (G=20) | 0.0030 ± 0.0023 | 0.0076 ± 0.0087 | 0.0011 ± 0.0011 | 0.0139 ± 0.0104 | 0.0139 ± 0.0087 | 0.0035 ± 0.0016 | 0.0059 ± 0.0051 | 0.0085 ± 0.0057 |
| PLCP (G=50) | 0.0022 ± 0.0019 | 0.0038 ± 0.0062 | 0.0011 ± 0.0014 | 0.0095 ± 0.0071 | 0.0077 ± 0.0094 | 0.0039 ± 0.0026 | 0.0031 ± 0.0014 | 0.0055 ± 0.0022 |
| Gaussian-Scoring | 0.0018 ± 0.0009 | 0.0024 ± 0.0009 | 0.0005 ± 0.0005 | 0.0127 ± 0.0130 | 0.0136 ± 0.0145 | 0.0024 ± 0.0012 | 0.0028 ± 0.0033 | 0.0079 ± 0.0035 |
| CQR | 0.0016 ± 0.0013 | 0.0024 ± 0.0009 | 0.0005 ± 0.0003 | 0.0104 ± 0.0069 | 0.0044 ± 0.0023 | 0.0007 ± 0.0014 | 0.0048 ± 0.0032 | 0.0084 ± 0.0011 |
| CQR-ALD | 0.0011 ± 0.0010 | 0.0016 ± 0.0017 | 0.0006 ± 0.0004 | 0.0049 ± 0.0075 | 0.0032 ± 0.0035 | 0.0013 ± 0.0016 | 0.0040 ± 0.0027 | 0.0125 ± 0.0025 |
| RCP | 0.0006 ± 0.0006 | 0.0029 ± 0.0011 | 0.0004 ± 0.0004 | 0.0023 ± 0.0020 | 0.0016 ± 0.0009 | 0.0014 ± 0.0015 | 0.0018 ± 0.0008 | 0.0048 ± 0.0011 |
| RCP-ALD | 0.0013 ± 0.0016 | 0.0022 ± 0.0011 | 0.0005 ± 0.0004 | 0.0072 ± 0.0110 | 0.0023 ± 0.0027 | 0.0012 ± 0.0010 | 0.0023 ± 0.0026 | 0.0039 ± 0.0013 |
| RCP-MultiHead | 0.0009 ± 0.0006 | 0.0036 ± 0.0011 | 0.0005 ± 0.0005 | 0.0019 ± 0.0012 | 0.0026 ± 0.0007 | 0.0012 ± 0.0009 | 0.0018 ± 0.0009 | 0.0045 ± 0.0016 |
| CPCP | 0.0006 ± 0.0006 | 0.0022 ± 0.0012 | 0.0003 ± 0.0002 | 0.0011 ± 0.0009 | 0.0004 ± 0.0003 | 0.0026 ± 0.0029 | 0.0015 ± 0.0007 | 0.0060 ± 0.0017 |
| CPCP (Clip) | 0.0008 ± 0.0006 | 0.0011 ± 0.0006 | 0.0003 ± 0.0003 | 0.0011 ± 0.0006 | 0.0005 ± 0.0003 | 0.0014 ± 0.0007 | 0.0010 ± 0.0005 | 0.0042 ± 0.0013 |
| CPCP (Mix) | 0.0005 ± 0.0005 | 0.0013 ± 0.0008 | 0.0003 ± 0.0002 | **0.0006 ± 0.0005** | **0.0003 ± 0.0002** | 0.0018 ± 0.0027 | 0.0008 ± 0.0005 | 0.0033 ± 0.0016 |
| CPCP (Clip+Mix) | 0.0004 ± 0.0005 | **0.0007 ± 0.0003** | **0.0002 ± 0.0002** | 0.0012 ± 0.0010 | **0.0003 ± 0.0003** | 0.0006 ± 0.0007 | **0.0005 ± 0.0003** | 0.0026 ± 0.0008 |
| CPCP-0.01 | 0.0007 ± 0.0006 | 0.0024 ± 0.0013 | 0.0005 ± 0.0004 | 0.0014 ± 0.0010 | **0.0003 ± 0.0003** | 0.0020 ± 0.0016 | 0.0015 ± 0.0008 | 0.0053 ± 0.0014 |
| CPCP-0.01 (Clip+Mix) | **0.0003 ± 0.0004** | 0.0009 ± 0.0007 | **0.0002 ± 0.0002** | 0.0007 ± 0.0005 | **0.0003 ± 0.0003** | 0.0005 ± 0.0008 | **0.0005 ± 0.0002** | 0.0033 ± 0.0012 |
| CPCP-0.05 | 0.0006 ± 0.0005 | 0.0022 ± 0.0017 | 0.0003 ± 0.0002 | 0.0009 ± 0.0006 | 0.0004 ± 0.0002 | 0.0018 ± 0.0014 | 0.0018 ± 0.0009 | 0.0053 ± 0.0027 |
| CPCP-0.05 (Clip+Mix) | **0.0003 ± 0.0005** | 0.0010 ± 0.0004 | **0.0002 ± 0.0002** | 0.0009 ± 0.0008 | **0.0003 ± 0.0002** | **0.0004 ± 0.0007** | 0.0005 ± 0.0003 | **0.0019 ± 0.0009** |

*Table 11.* Full comparison of Volume. Volume ($\downarrow$) on all benchmarks with all baselines.

| Method | Bike | Diamond | Gas Turbine | Naval | SGEMM | Superconductivity | Transcoding | WEC |
|---|---|---|---|---|---|---|---|---|
| Split | 0.7661 ± 0.0222 | 0.3793 ± 0.0158 | 0.1794 ± 0.0202 | -0.2741 ± 0.2199 | -1.7916 ± 0.0618 | 0.9917 ± 0.0408 | -1.0367 ± 0.0727 | 1.3004 ± 0.0207 |
| PLCP (G=20) | 0.7528 ± 0.0303 | 0.3520 ± 0.0149 | 0.1468 ± 0.0253 | -0.7786 ± 0.2926 | -1.8795 ± 0.1089 | 0.8815 ± 0.0325 | -1.3365 ± 0.1637 | 1.0509 ± 0.1180 |
| PLCP (G=50) | 0.7419 ± 0.0571 | 0.3427 ± 0.0147 | 0.1529 ± 0.0305 | -0.7513 ± 0.3553 | -1.9906 ± 0.1200 | 0.8848 ± 0.0349 | -1.4444 ± 0.0921 | 1.0260 ± 0.0843 |
| Gaussian-Scoring | **-0.7455 ± 0.0498** | **-1.6312 ± 0.0612** | **0.0454 ± 0.0159** | 1.1504 ± 0.6506 | 1.1303 ± 1.5991 | **-0.6082 ± 0.0486** | -1.1826 ± 0.7100 | 1.6395 ± 0.2775 |
| CQR | 0.8118 ± 0.0441 | 0.2990 ± 0.0084 | 0.0603 ± 0.0181 | 0.0665 ± 0.3057 | -2.3191 ± 0.0515 | 0.8934 ± 0.0372 | **-2.4395 ± 0.1315** | **0.6992 ± 0.0189** |
| CQR-ALD | 1.0122 ± 0.0889 | 0.3170 ± 0.0344 | 0.0524 ± 0.0187 | 0.6268 ± 0.2757 | **-2.4332 ± 0.0748** | 0.9521 ± 0.0495 | -2.2226 ± 0.1615 | 0.9541 ± 0.0243 |
| RCP | 0.7402 ± 0.0638 | 0.3197 ± 0.0126 | 0.1580 ± 0.0288 | -0.9301 ± 0.2656 | -2.0543 ± 0.0667 | 0.9009 ± 0.0842 | -1.5827 ± 0.0699 | 0.7489 ± 0.0249 |
| RCP-ALD | 0.7380 ± 0.0415 | 0.3219 ± 0.0125 | 0.1406 ± 0.0227 | -0.8320 ± 0.3071 | -2.0520 ± 0.0795 | 0.8611 ± 0.0318 | -1.5375 ± 0.0824 | 0.7571 ± 0.0292 |
| RCP-MultiHead | 0.7339 ± 0.0269 | 0.3156 ± 0.0080 | 0.1546 ± 0.0242 | -1.0368 ± 0.2576 | -2.0542 ± 0.0608 | 0.9002 ± 0.0713 | -1.5889 ± 0.0894 | 0.7492 ± 0.0258 |
| CPCP | 0.7423 ± 0.0468 | 0.3723 ± 0.0420 | 0.1456 ± 0.0386 | -0.9894 ± 0.3102 | -2.0635 ± 0.0782 | 0.9247 ± 0.1059 | -1.4728 ± 0.1361 | 0.9208 ± 0.0541 |
| CPCP (Clip) | 0.7512 ± 0.0532 | 0.3459 ± 0.0137 | 0.1555 ± 0.0334 | -0.8944 ± 0.3215 | -2.0523 ± 0.0568 | 0.8767 ± 0.0333 | -1.5072 ± 0.0840 | 0.8472 ± 0.0457 |
| CPCP (Mix) | 0.7370 ± 0.0433 | 0.3566 ± 0.0317 | 0.1380 ± 0.0308 | -1.0269 ± 0.2389 | -2.0764 ± 0.0482 | 0.9123 ± 0.0669 | -1.5034 ± 0.1004 | 0.8262 ± 0.0469 |
| CPCP (Clip+Mix) | 0.7486 ± 0.0330 | 0.3381 ± 0.0120 | 0.1511 ± 0.0423 | -0.9101 ± 0.1810 | -2.0793 ± 0.0402 | 0.8778 ± 0.0236 | -1.5591 ± 0.0597 | 0.7697 ± 0.0461 |
| CPCP-0.01 | 0.7483 ± 0.0492 | 0.3928 ± 0.1133 | 0.1582 ± 0.0478 | -0.9517 ± 0.2808 | -2.0613 ± 0.0403 | 0.9197 ± 0.0769 | -1.4551 ± 0.0713 | 0.9045 ± 0.0804 |
| CPCP-0.01 (Clip+Mix) | 0.7553 ± 0.0373 | 0.3437 ± 0.0132 | 0.1501 ± 0.0358 | -0.9692 ± 0.2858 | -2.0506 ± 0.0754 | 0.8669 ± 0.0286 | -1.5704 ± 0.0712 | 0.7614 ± 0.0477 |
| CPCP-0.05 | 0.7471 ± 0.0400 | 0.4128 ± 0.1902 | 0.1439 ± 0.0343 | **-1.0389 ± 0.2635** | -2.0583 ± 0.0536 | 0.8895 ± 0.0424 | -1.4253 ± 0.1310 | 0.8521 ± 0.1029 |
| CPCP-0.05 (Clip+Mix) | 0.7473 ± 0.0445 | 0.3340 ± 0.0118 | 0.1347 ± 0.0244 | -1.0265 ± 0.2652 | -2.0449 ± 0.0639 | 0.8724 ± 0.0329 | -1.5644 ± 0.0611 | 0.7037 ± 0.0349 |

## B. Proofs

### B.1. Proof of Proposition 3.1

We fix $x$ throughout the proof and suppress the dependence on $x$ for notational simplicity. Let $F(\cdot) = F_{S|X=x}(\cdot)$ denote the conditional cumulative distribution function of the score $S = s(x, Y)$, and let

$$\mathcal{L}(q) = \mathbb{E}\big[\rho_\tau(q, S) \mid X = x\big]$$

denote the corresponding expected pinball loss. Let $q^\star = q_\tau(x)$ be the true $\tau$-quantile, i.e., $F(q^\star) = \tau$, and let $\hat{q} = \hat{q}_\tau(x)$.

It is well known that $\mathcal{L}(\cdot)$ is a convex and absolutely continuous function of $q$. Moreover, for almost every $q \in \mathbb{R}$,

$$\frac{d}{dq}\mathcal{L}(q) = F(q) - \tau. \tag{39}$$

Since $q^\star$ is a minimizer of $\mathcal{L}$, we have $F(q^\star) - \tau = 0$.

By the fundamental theorem of calculus for absolutely continuous functions, the excess risk can therefore be written as

$$\mathcal{L}(\hat{q}) - \mathcal{L}(q^\star) = \int_{q^\star}^{\hat{q}} \big(F(z) - \tau\big) dz. \tag{40}$$

We now use the Lipschitz assumption on the conditional CDF. Since $F$ is $L_F$-Lipschitz, it is almost everywhere differentiable with density $f(z) = F'(z)$ satisfying $f(z) \le L_F$ for almost every $z$.

Without loss of generality, assume $\hat{q} \ge q^\star$ (the case $\hat{q} < q^\star$ follows by symmetry). Define $g(z) = F(z) - \tau$, so that $g(q^\star) = 0$, $g(z) \ge 0$ for $z \ge q^\star$, and $g'(z) = f(z) \le L_F$ almost everywhere.

Consider the auxiliary function

$$H(q) = \int_{q^\star}^{q} g(z) dz - \frac{1}{2L_F} g(q)^2, \qquad q \ge q^\star.$$

For almost every $q \ge q^\star$, we have

$$H'(q) = g(q)\Big(1 - \frac{g'(q)}{L_F}\Big) = g(q)\Big(1 - \frac{f(q)}{L_F}\Big) \ge 0,$$

where the inequality follows from $g(q) \ge 0$ and $f(q) \le L_F$. Since $H(q^\star) = 0$ and $H$ is non-decreasing on $[q^\star, \infty)$, it follows that $H(\hat{q}) \ge 0$, i.e.,

$$\int_{q^\star}^{\hat{q}} \big(F(z) - \tau\big) dz \ge \frac{1}{2L_F}\big(F(\hat{q}) - \tau\big)^2. \tag{41}$$

Combining (40) and (41) yields

$$\mathcal{L}(\hat{q}) - \mathcal{L}(q^\star) \ge \frac{1}{2L_F}\big(F(\hat{q}) - \tau\big)^2,$$

which implies

$$\big|F(\hat{q}) - \tau\big| \le \sqrt{2L_F\big(\mathcal{L}(\hat{q}) - \mathcal{L}(q^\star)\big)}.$$

This completes the proof. $\qquad\square$

### B.2. Proof of Proposition 3.4

We aim to characterize the approximation error between the squared conditional coverage mismatch and the density-weighted excess risk. Let $\epsilon_q(x) := \hat{q}_\tau(x) - q_\tau(x)$ denote the estimation error.

**Expansion of the Squared Coverage Error.** Let $G(u) := (F_{S|X}(u) - \tau)^2$. We seek the Taylor expansion of $G(\hat{q}_\tau(x))$ around the true quantile $q_\tau(x)$. Note that $F_{S|X}(q_\tau(x)) = \tau$. We compute the derivatives of $G(u)$ evaluated at $u = q_\tau(x)$:

- First derivative:
$$G'(u) = 2(F_{S|X}(u) - \tau)f_{S|X}(u) \implies G'(q_\tau(x)) = 0. \tag{42}$$

- Second derivative:
$$G''(u) = 2f_{S|X}(u)^2 + 2(F_{S|X}(u) - \tau)f'_{S|X}(u) \implies G''(q_\tau(x)) = 2f_{S|X}(q_\tau(x))^2. \tag{43}$$

- Third derivative:
$$G'''(u) = 6f_{S|X}(u)f'_{S|X}(u) + 2(F_{S|X}(u) - \tau)f''_{S|X}(u). \tag{44}$$

Applying Taylor expansion with the Lagrange remainder to the third order:

$$
\begin{aligned}
(F_{S|X}(\hat{q}_\tau(x)) - \tau)^2 &= G(q_\tau(x)) + G'(q_\tau(x))\epsilon_q(x) + \frac{1}{2}G''(q_\tau(x))\epsilon_q(x)^2 + \frac{1}{6}G'''(\xi_{S,1})\epsilon_q(x)^3 \\
&= f_{S|X}(q_\tau(x))^2\epsilon_q(x)^2 + \frac{1}{6}G'''(\xi_{S,1})\epsilon_q(x)^3,
\end{aligned}
\tag{45}
$$

where $\xi_{S,1}$ lies between $\hat{q}_\tau(x)$ and $q_\tau(x)$.

**Expansion of the Excess Risk.** Recall the excess risk $\mathcal{E}(x) := \mathcal{L}_x(\hat{q}_\tau(x)) - \mathcal{L}_x(q_\tau(x))$, where $\mathcal{L}_x(\cdot)$ is the expected pinball loss. The derivatives of $\mathcal{L}_x$ at $q_\tau(x)$ are:

- $\mathcal{L}'_x(q_\tau(x)) = F_{S|X}(q_\tau(x)) - \tau = 0.$
- $\mathcal{L}''_x(q_\tau(x)) = f_{S|X}(q_\tau(x)).$
- $\mathcal{L}'''_x(u) = f'_{S|X}(u).$

Applying Taylor's theorem to $\mathcal{E}(x)$ up to the third order:

$$
\begin{aligned}
\mathcal{E}(x) &= \frac{1}{2}\mathcal{L}''_x(q_\tau(x))\epsilon_q(x)^2 + \frac{1}{6}\mathcal{L}'''_x(\xi_{S,2})\epsilon_q(x)^3 \\
&= \frac{1}{2}f_{S|X}(q_\tau(x))\epsilon_q(x)^2 + \frac{1}{6}f'_{S|X}(\xi_{S,2})\epsilon_q(x)^3,
\end{aligned}
\tag{46}
$$

where $\xi_{S,2}$ lies between $\hat{q}_\tau(x)$ and $q_\tau(x)$.

Multiplying by $2f_{S|X}(q_\tau(x))$ to match the leading term of the squared coverage error:

$$2f_{S|X}(q_\tau(x))\mathcal{E}(x) = f_{S|X}(q_\tau(x))^2\epsilon_q(x)^2 + \frac{1}{3}f_{S|X}(q_\tau(x))f'_{S|X}(\xi_{S,2})\epsilon_q(x)^3. \tag{47}$$

**Characterizing the Approximation Gap.** Subtracting the weighted excess risk from the squared coverage error, the second-order terms cancel out perfectly:

$$
\begin{aligned}
&\left(F_{S|X}(\hat{q}_\tau(x)) - \tau\right)^2 - 2f_{S|X}(q_\tau(x))\mathcal{E}(x) \\
&= \left[f_{S|X}(q_\tau(x))^2\epsilon_q(x)^2 + \frac{1}{6}G'''(\xi_{S,1})\epsilon_q(x)^3\right] - \left[f_{S|X}(q_\tau(x))^2\epsilon_q(x)^2 + \frac{1}{3}f_{S|X}(q_\tau(x))f'_{S|X}(\xi_{S,2})\epsilon_q(x)^3\right] \\
&= \frac{1}{6}\epsilon_q(x)^3 \left[G'''(\xi_{S,1}) - 2f_{S|X}(q_\tau(x))f'_{S|X}(\xi_{S,2})\right].
\end{aligned}
\tag{48}
$$

Substituting the expression for $G'''(\xi_{S,1})$:

$$\mathrm{Gap}(x) = \frac{1}{6}\epsilon_q(x)^3 \left[6f(\xi_{S,1})f'(\xi_{S,1}) + 2(F(\xi_{S,1}) - \tau)f''(\xi_{S,1}) - 2f(q_\tau(x))f'(\xi_{S,2})\right], \tag{49}$$

where we abbreviated $f_{S|X}$ as $f$ and $F_{S|X}$ as $F$.

We now introduce the necessary regularity assumptions as follows.

**Assumption B.1** (Regularity assumption on smoothness of $f_{S|X}$)**.** We assume that $f_{S|X}$ is twice continuous differentiable, and there exist constants $B_w, B_f, B_{f'}, B_{f''}$ such that:

$$|f(q(x))| \leq B_w, \quad |f(\xi)| \leq B_f, \quad |f'(\xi)| \leq B_{f'}, \quad |f''(\xi)| \leq B_{f''}, \tag{50}$$

such that the following bounds hold for each $x \in \mathcal{X}, \xi \in \mathcal{S}$, where $\mathcal{S}$ is the space of nonconformity scores.

Here, we add a refined constant $B_w$ because it is the bound on true weights that will appear again in Theorem 5.2.

With Assumption B.1, we have:

$$\left| \left( F_{S|X}(\hat{q}_\tau(x)) - \tau \right)^2 - 2 f_{S|X}(q_\tau(x)) \mathcal{E}(x) \right| \leq C_f \epsilon_q(x)^3,$$

where

$$C_f = (6 B_f B_{f'} + 2 \max\{\tau, 1 - \tau\} B_{f''} + 2 B_w B_{f'}).$$

This confirms that minimizing the density-weighted pinball loss is a highly accurate surrogate for minimizing the MSQE, with a negligible approximation gap in the regime of consistent quantile estimation.

□

## B.3. Proof of Theorem 5.2

### B.3.1. Preliminary: fast rates via local Rademacher complexity

We emphasize that Assumption 5.1 in the main text isolates only the geometric and smoothness conditions that determine the constants appearing in the excess risk bound. The fast convergence of the auxiliary quantile estimators is established separately via standard local Rademacher complexity arguments, which we briefly summarize below for completeness.

**Auxiliary quantile estimation error.** Recall that the quality of the estimated weights depends on the accuracy of the auxiliary quantile estimators. Define

$$\mathcal{E}_q(n) := \sup_{\beta \in \{\tau - \delta, \tau + \delta\}} \left\| \hat{q}_\beta - q_\beta \right\|_{L_2(\mathbb{P}_X)}.$$

Without additional curvature assumptions, global complexity bounds typically yield the slow rate $\mathcal{E}_q(n) = O_{\mathbb{P}}(n^{-1/4})$. To obtain fast rates, we invoke the theory of local Rademacher complexity (Bartlett et al., 2005).

**Localization.** For $r > 0$, define the localized hypothesis class

$$\mathcal{G}_r := \left\{ q \in \mathcal{G} : \|q - q^\star\|_{L_2(\mathbb{P}_X)} \leq r \right\}.$$

All subsequent arguments are restricted to $\mathcal{G}_r$, where $r$ is chosen according to the fixed-point equation associated with the local Rademacher complexity.

**Bernstein condition.** Fast-rate results require the loss to satisfy a Bernstein (variance) condition. This property follows from the curvature of the pinball risk.

1. **Quadratic growth.** By Assumption 5.1 (density lower bound), there exists $b_w > 0$ such that $f_{S|X}(q_\tau(x) \mid x) \geq b_w$ for all $x \in \mathcal{X}$. Standard arguments for quantile regression imply that the expected pinball risk satisfies the local quadratic growth condition

$$\mathcal{R}_{\text{pin}}(q) - \mathcal{R}_{\text{pin}}(q^\star) \geq \frac{b_w}{2} \|q - q^\star\|_{L_2(\mathbb{P}_X)}^2, \qquad \forall q \in \mathcal{G}_r.$$

2. **Variance control.** The pinball loss $\rho_\tau$ is $L_\rho$-Lipschitz in its prediction argument, with $L_\rho = \max(\tau, 1 - \tau)$. Consequently,

$$\left| \rho_\tau(q, S) - \rho_\tau(q^\star, S) \right| \leq L_\rho |q - q^\star|.$$

This yields

$$\mathrm{Var}(\rho_\tau(q, S) - \rho_\tau(q^\star, S)) \leq L_\rho^2 \|q - q^\star\|_{L_2(\mathbb{P}_X)}^2.$$

Combining with the quadratic growth inequality gives the Bernstein condition

$$\mathrm{Var}(\rho_\tau(q, S) - \rho_\tau(q^\star, S)) \leq B_{\mathrm{var}}\big(\mathcal{R}_{\mathrm{pin}}(q) - \mathcal{R}_{\mathrm{pin}}(q^\star)\big), \qquad B_{\mathrm{var}} = \frac{2L_\rho^2}{b_w}. \tag{51}$$

**Fast-rate consequence.** Under the Bernstein condition (51) and standard sub-root assumptions on the local Rademacher complexity (e.g., $\mathfrak{R}_n(\mathcal{G}_r) \lesssim r\sqrt{d/n}$ for VC-type classes), the fixed-point equation

$$\mathfrak{R}_n(\mathcal{G}_{r^\star}) \asymp r^{\star 2}$$

yields $r^\star \asymp \sqrt{d/n}$. As a result, the ERM-based auxiliary quantile estimators satisfy

$$\mathcal{E}_q(n) = \|\hat{q}_\beta - q_\beta\|_{L_2(\mathbb{P}_X)} \leq C_{\mathrm{fast}}\mathfrak{R}_n(\mathcal{G}) = O_\mathbb{P}(n^{-1/2}),$$

uniformly over $\beta \in \{\tau - \delta, \tau + \delta\}$.

### B.3.2. ERROR DECOMPOSITION

We define the following empirical risks:

- $\mathcal{R}_n(g) := \frac{1}{n}\sum_{i=1}^n w(x_i)\rho_\tau(g(x_i), s_i)$ denotes the empirical risk with *true* weights $w$.

- $\hat{\mathcal{R}}_n(g) := \frac{1}{n}\sum_{i=1}^n \hat{w}(x_i)\rho_\tau(g(x_i), s_i)$ denotes the empirical risk with *estimated* weights $\hat{w}$.

Using a standard empirical-process argument, we control the excess risk as:

$$\mathcal{R}(\hat{g}) - \mathcal{R}(g^\star) \leq 2\sup_{g \in \mathcal{G}} |\hat{\mathcal{R}}_n(g) - \mathcal{R}(g)| \tag{52}$$

$$\leq \underbrace{2\sup_{g \in \mathcal{G}} |\mathcal{R}_n(g) - \mathcal{R}(g)|}_{\text{Term I}} + \underbrace{2\sup_{g \in \mathcal{G}} |\hat{\mathcal{R}}_n(g) - \mathcal{R}_n(g)|}_{\text{Term II}}. \tag{53}$$

### B.3.3. BOUNDING THE STOCHASTIC ERROR (TERM I)

Term I corresponds to the generalization error with fixed (oracle) weights. To avoid imposing a uniform boundedness assumption on the loss, we introduce the following conditional sub-Gaussian assumption.

**Assumption B.2** (Conditional sub-Gaussian noise). The nonconformity score $S$ is conditionally sub-Gaussian given $X$. Specifically, there exists a constant $\sigma_S > 0$ such that for all $\lambda \in \mathbb{R}$,

$$\mathbb{E}\big[\exp\big(\lambda\big(S - \mathbb{E}[S \mid X]\big)\big) \mid X\big] \leq \exp\left(\frac{\sigma_S^2\lambda^2}{2}\right) \quad \text{a.s.} \tag{54}$$

By Assumption 5.1.3, the weight function satisfies $w(x) \leq B_w$ almost surely. Moreover, the pinball loss $\rho_\tau$ is $L_\rho$-Lipschitz in its prediction argument, where $L_\rho = \max(\tau, 1 - \tau)$. Therefore, the weighted loss

$$(x, s) \mapsto w(x)\rho_\tau(g(x), s)$$

is $B_w L_\rho$-Lipschitz with respect to the prediction argument $g(x)$.

Under the conditional sub-Gaussian assumption in Assumption B.2, the random variable $w(X)\rho_\tau(g(X), S)$ is conditionally sub-exponential with parameter $B_w L_\rho \sigma_S$. Applying standard global Rademacher complexity bounds together with Bernstein-type concentration inequalities yields that, with probability at least $1 - \zeta$,

$$\text{Term I} \leq 2B_w L_\rho \mathfrak{R}_n(\mathcal{G}) + B_w L_\rho \sigma_S\sqrt{\frac{2\log(1/\zeta)}{n}}. \tag{55}$$

### B.3.4. BOUNDING THE WEIGHT ESTIMATION ERROR (TERM II)

This term captures the error arising from using estimated weights $\hat{w}$ instead of true weights $w$. We bound this difference uniformly over the hypothesis class $\mathcal{G}$ by applying the Cauchy-Schwarz inequality. The derivation proceeds as follows:

$$
\begin{aligned}
\text{Term II} &:= \sup_{g \in \mathcal{G}} |\hat{\mathcal{R}}_n(g) - \mathcal{R}_n(g)| \\
&= \sup_{g \in \mathcal{G}} \left| \frac{1}{n} \sum_{i=1}^n \hat{w}(x_i) \rho_\tau(g(x_i), s_i) - \frac{1}{n} \sum_{i=1}^n w(x_i) \rho_\tau(g(x_i), s_i) \right| \\
&= \sup_{g \in \mathcal{G}} \left| \frac{1}{n} \sum_{i=1}^n (\hat{w}(x_i) - w(x_i)) \rho_\tau(g(x_i), s_i) \right| \\
&\leq \sup_{g \in \mathcal{G}} \left( \sqrt{\frac{1}{n} \sum_{i=1}^n (\hat{w}(x_i) - w(x_i))^2} \cdot \sqrt{\frac{1}{n} \sum_{i=1}^n \rho_\tau(g(x_i), s_i)^2} \right) \\
&= \left( \frac{1}{n} \sum_{i=1}^n (\hat{w}(x_i) - w(x_i))^2 \right)^{1/2} \cdot \sup_{g \in \mathcal{G}} \left( \frac{1}{n} \sum_{i=1}^n \rho_\tau(g(x_i), s_i)^2 \right)^{1/2} \\
&= \|\hat{w} - w\|_{2,n} \cdot M_{\rho, \mathcal{G}},
\end{aligned}
\tag{56}
$$

where $M_{\rho, \mathcal{G}} := \sup_{g \in \mathcal{G}} (\mathbb{E}_n[\rho_\tau(g(X), S)^2])^{1/2}$ represents the maximal empirical root-mean-square (RMS) of the loss function over the hypothesis class, and $\| \cdot \|_{2,n}$ represents the empirical $L_2$-norm.

For notational convenience, we define:

$$
\begin{aligned}
\Delta_\delta(x) &:= \frac{q_{\tau+\delta}(x) - q_{\tau-\delta}(x)}{2\delta} \\
\hat{\Delta}_\delta(x) &:= \frac{\hat{q}_{\tau+\delta}(x) - \hat{q}_{\tau-\delta}(x)}{2\delta},
\end{aligned}
\tag{57}
$$

and

$$
w_\delta(x) := \frac{1}{\Delta_\delta(x)}
\tag{58}
$$

as the population finite-difference weight approximation.

We decompose the weight RMSE $\|\hat{w} - w\|_{L_2}$ into bias (finite difference approximation) and variance (estimation error):

$$
\|\hat{w} - w\|_{2,n} \leq \|w_\delta - w\|_{2,n} + \|\hat{w} - w_\delta\|_{2,n}.
\tag{59}
$$

**Bias analysis.** We aim to bound the bias $\|w_\delta - w\|_{2,n}$ introduced by the central finite-difference approximation pointwise. Recall that the target weight is the reciprocal of the quantile density function (sparsity): $w(x) = 1/q_\tau'(x)$.

First, we analyze the error of the central difference estimator $\Delta_\delta(x)$. Fix $x$ and perform a Taylor expansion with Lagrange remainder of the quantile function $q_\beta(x)$ with respect to $\beta$ around $\tau$:

$$
q_{\tau+\delta}(x) = q_\tau(x) + \delta q_\tau'(x) + \frac{\delta^2}{2} q_\tau''(x) + \frac{\delta^3}{6} q_{\xi_1}'''(x),
\tag{60}
$$

$$
q_{\tau-\delta}(x) = q_\tau(x) - \delta q_\tau'(x) + \frac{\delta^2}{2} q_\tau''(x) - \frac{\delta^3}{6} q_{\xi_2}'''(x),
\tag{61}
$$

where $\xi_1 \in (\tau, \tau+\delta)$ and $\xi_2 \in (\tau-\delta, \tau)$. We clarify that we fix $x$ and use $q_\tau'(x)$ to denote the derivative of $q_\tau(x)$ w.r.t. $\tau$.

Subtracting the two equations cancels the quadratic terms (second derivatives), yielding:

$$
q_{\tau+\delta}(x) - q_{\tau-\delta}(x) = 2\delta q_\tau'(x) + \frac{\delta^3}{6}(q_{\xi_1}'''(x) + q_{\xi_2}'''(x)).
\tag{62}
$$

Dividing by $2\delta$, the error in the derivative estimation is bounded by the third derivative:

$$
|\Delta_\delta(x) - q_\tau'(x)| = \left| \frac{\delta^2}{12}(q_{\xi_1}'''(x) + q_{\xi_2}'''(x)) \right| \leq \frac{\delta^2}{6} \sup_\beta |q_\beta'''(x)| \leq \frac{B_q'''}{6} \delta^2.
\tag{63}
$$

Here, we used Assumption 5.1 (1) which bounds the third derivative of the quantile function by $B_q'''$.

Next, we leverage Assumption 5.1 (2) ($1/q_\tau'(x) \le B_w$) and to make the discussion meaningful, we must assume $\delta$ is small enough ($\delta \le \sqrt{3/B_w B_q'''}$), so that:

$$\Delta_\delta(x) \ge q_\tau'(x) - |\Delta_\delta(x) - q_\tau'(x)| \ge \frac{1}{B_w} - \frac{B_q'''}{6}\delta^2 \ge \frac{1}{2B_w} \tag{64}$$

Therefore, we have

$$\begin{aligned}
|w_\delta(x) - w(x)| &= \frac{|q_\tau'(x) - \Delta_\delta(x)|}{\Delta_\delta(x)q_\tau'(x)} \\
&\le \frac{B_w^2 B_q''' \delta^2}{3},
\end{aligned} \tag{65}$$

and naturally,

$$\|w_\delta - w\|_{2,n} \le \frac{B_w^2 B_q''' \delta^2}{3}. \tag{66}$$

**Variance analysis.** We aim to bound the variance term, $\|\hat{w} - w_\delta\|_{2,n}$. The problem that cannot be steered around when using estimated reciprocal weights is the uniform lower bound for $\hat{\Delta}_\delta(x)$, where the uniform range is both for $x \in \mathcal{X}$ and function $\hat{q} \in \mathcal{G}$. Here, we try to weaken the needed conditions and utilize the assumption on relation between infinity norm and $L_2$-norm, locally.

**Assumption B.3** (Norm equivalence via Hölder regularity). Let $\mathcal{X} \subset \mathbb{R}^d$ be a compact domain with nonempty interior, and let $P_X$ be a distribution supported on $\mathcal{X}$ with density bounded away from zero and infinity. Assume that there exists a smoothness index $s > 0$ and a constant $R > 0$ such that for all $g \in \mathcal{G}_r$, the difference $g - g^\star$ belongs to the Hölder class $C^s(\mathcal{X})$ with

$$\|g - g^\star\|_{C^s(\mathcal{X})} \le R.$$

Then there exists a constant $C_H > 0$, depending only on $(s, d, \mathcal{X}, P_X)$, such that for all $g \in \mathcal{G}_r$,

$$\|g - g^\star\|_{L_\infty(\mathcal{X})} \le C_H R^{\frac{d}{2s+d}} \|g - g^\star\|_{L_2(P_X)}^{\frac{2s}{2s+d}}. \tag{67}$$

We note that we should consider two $g^\star$s, i.e., $q_\beta$ for $\beta \in \{\tau \pm \delta\}$. For notational convenience, let $(C_{\text{norm}}, \nu)$ be the constants in Assumption 5.1. When Assumption 5.1.3 is verified via the Hölder-type norm interpolation in Lemma B.6, one may take $C_{\text{norm}} = C_H R^{d/(2s+d)}$ and $\nu_H = 2s/(2s+d)$. To avoid confusion, we keep $\nu$ for the abstract exponent in Assumption 5.1.3 and use $\nu_H$ only for this Hölder instantiation.

*Remark* B.4. For classic statistical regression models, e.g., parametric model or Reproducing Kernel Hilbert Space (RKHS) regression with smooth kernel (e.g., Gaussian kernel), we have $\nu \approx 1$.

With Assumption B.3 at hand, we can construct the uniform lower bound on $\hat{w}(x)$ with probability at least $1 - 2\zeta$:

$$\begin{aligned}
\hat{\Delta}_\delta(x) &\ge \Delta_\delta(x) - |\Delta_\delta(x) - \hat{\Delta}_\delta(x)| \\
&\ge \Delta_\delta(x) - \frac{1}{2\delta}(|\hat{q}_{\tau+\delta}(x) - q_{\tau+\delta}(x)| + |\hat{q}_{\tau-\delta}(x) - q_{\tau-\delta}(x)|) \\
&\ge \frac{1}{2B_w} - \frac{1}{2\delta}(\|\hat{q}_{\tau+\delta} - q_{\tau+\delta}\|_{L_\infty} + \|\hat{q}_{\tau-\delta} - q_{\tau-\delta}\|_{L_\infty}) \\
&\ge \frac{1}{4B_w},
\end{aligned} \tag{68}$$

For the last inequality, we actually need:

$$C_{\text{norm}} (C_{\text{fast}} \mathfrak{R}_n(\mathcal{G}))^\nu \le \frac{\delta}{4B_w}. \tag{69}$$

In fact, our optimal rate is given by $\delta^\star \asymp \mathfrak{R}_n(\mathcal{G})^{1/3}$, and thus we need $\nu \ge 1/3$, otherwise this stability condition would induce a slower rate. In the Hölder instantiation of Lemma B.6 where $\nu = \nu_H = 2s/(2s + d)$, the condition $\nu \ge 1/3$ is

equivalent to $d \leq 4s$, reflecting the standard curse of dimensionality. We remark that this bottleneck motivates our clipping mechanism on the weights.

Thus, we have:

$$\|\hat{w} - w_\delta\|_{2,n} = \left\| \frac{\Delta_\delta - \hat{\Delta}_\delta}{\hat{\Delta}_\delta \Delta_\delta} \right\|_{2,n} \leq 8B_w^2 \|\hat{\Delta}_\delta - \Delta_\delta\|_{2,n} \tag{70}$$

To avoid a rate reduction, instead of directly bounding the gap through Assumption B.3, we choose to transform the empirical norm to population norm using a concentration inequality. Specifically, we define the random variable:

$$Z(X) = \left( 2\delta(\hat{\Delta}_\delta(X) - \Delta_\delta(X)) \right)^2 . \tag{71}$$

Since we consider $\hat{q}_\beta \in \mathcal{G}$, we can apply Assumption B.3 and control the variance of $Z$ as:

$$\text{Var}(Z) \leq \mathbb{E}[Z^2] \leq \|Z\|_{L_\infty(\mathcal{X})} \mathbb{E}[Z] \leq C_{\text{norm}}^2 \left\| \hat{\Delta}_\delta - \Delta_\delta \right\|_{L_2}^{2+2\nu} . \tag{72}$$

Moreover, $Z$ is also bounded, because:

$$|Z(x)| \leq \|Z\|_{L_\infty(\mathcal{X})}^2 \leq C_{\text{norm}}^2 \left\| \hat{\Delta}_\delta - \Delta_\delta \right\|_{L_2}^{2\nu} \leq C_{\text{norm}}^2 \mathcal{E}_q(n)^{2\nu} \tag{73}$$

Hence, we can apply Bernstein's inequality (Bach, 2024) to transform the empirical norm into population norm, with probability at least $1 - \zeta$:

$$\|2\delta(\hat{\Delta}_\delta - \Delta_\delta)\|_{2,n}^2 \leq \|2\delta(\hat{\Delta}_\delta - \Delta_\delta)\|_{L_2}^2 + \sqrt{\frac{2\left( C_{\text{norm}}^2 \left\| \hat{\Delta}_\delta - \Delta_\delta \right\|_{L_2}^{2+2\nu} \right) \log(1/\zeta)}{n}} + \frac{2C_{\text{norm}}^2 \mathcal{E}_q(n)^{2\nu} \log(1/\zeta)}{3n} . \tag{74}$$

$$\leq 4\mathcal{E}_q(n)^2 + C_{\text{norm}} \mathcal{E}_q(n)^{1+\nu} \sqrt{\frac{2^{3+2\nu} \log(1/\zeta)}{n}} + \frac{2C_{\text{norm}}^2 \mathcal{E}_q(n)^{2\nu} \log(1/\zeta)}{3n} .$$

### B.3.5. OPTIMAL BANDWIDTH AND FINAL BOUND

Combining the bias and variance terms derived above, the error Term II can be written as $U(\delta)$:

$$U(\delta) = M_{\rho,\mathcal{G}}(c_1 \delta^2 + c_2 \delta^{-1}),$$

where

$$c_1 = \frac{B_w^2 B_q'''}{3}, \tag{75}$$

and

$$c_2 = 4B_w^2 \sqrt{4\mathcal{E}_q(n)^2 + C_{\text{norm}} \mathcal{E}_q(n)^{1+\nu} \sqrt{\frac{2^{3+2\nu} \log(1/\zeta)}{n}} + \frac{2C_{\text{norm}}^2 \mathcal{E}_q(n)^{2\nu} \log(1/\zeta)}{3n}} . \tag{76}$$

Minimizing $U(\delta)$ with respect to $\delta$ yields the optimal bandwidth

$$\delta^\star = \left( \frac{c_2}{2c_1} \right)^{1/3} .$$

Substituting $\delta^\star$ back into $U(\delta)$, we obtain

$$U(\delta^\star)/M_{\rho,\mathcal{G}} = 3 \left( \frac{c_1}{4} \right)^{1/3} c_2^{2/3}$$

$$= C_{\text{const}} B_w^2 (B_q''')^{1/3} \left( 4\mathcal{E}_q(n)^2 + C_{\text{norm}} \mathcal{E}_q(n)^{1+\nu} \sqrt{\frac{2^{3+2\nu} \log(1/\zeta)}{n}} + \frac{2C_{\text{norm}}^2 \mathcal{E}_q(n)^{2\nu} \log(1/\zeta)}{3n} \right)^{1/3} , \tag{77}$$

where $C_{\text{const}} = 2 \times 3^{2/3}$.

We now invoke the fast-rate guarantee on the local neighborhood $\mathcal{G}_r$. Specifically, under the local Rademacher complexity condition,

$$\mathcal{E}_q(n) = \|\hat{q}_\beta - q_\beta\|_{L_2(P_X)} \leq C_{\text{fast}}\mathfrak{R}_n(\mathcal{G}) = O(n^{-1/2}) \tag{78}$$

with probability at least $1 - \zeta$.

We conclude that

$$U(\delta^\star)/M_{\rho,\mathcal{G}} = O\left(\mathfrak{R}_n(\mathcal{G})^{2/3}\right),$$

and in particular,

$$U(\delta^\star) = O(n^{-1/3}),$$

with probability at least $1 - 2\zeta$.

Combining Term I and Term II, with probability at least $1 - 3\zeta$, we have the final bound on the generalization error $\mathcal{R}(\hat{g}) - \mathcal{R}(g^\star)$:

$$\mathcal{R}(\hat{g}) - \mathcal{R}(g^\star) \leq 2(\text{Term I} + \text{Term II})$$

$$\leq 4B_w L_\rho \mathfrak{R}_n(\mathcal{G}) + 2B_w L_\rho \sigma_S \sqrt{\frac{2\log(1/\zeta)}{n}}$$

$$+ 2M_{\rho,\mathcal{G}} C_{\text{const}} B_w^2 (B_q''')^{1/3} \left(4\mathcal{E}_q(n)^2 + C_{\text{norm}}\mathcal{E}_q(n)^{1+\nu}\sqrt{\frac{2^{3+2\nu}\log(1/\zeta)}{n}} + \frac{2C_{\text{norm}}^2\mathcal{E}_q(n)^{2\nu}\log(1/\zeta)}{3n}.\right)^{1/3} \tag{79}$$

It is easy to check that the weight estimation error is the leading term with appropriately large $n$. Hence, we conclude that:

$$\mathcal{R}(\hat{g}) - \mathcal{R}(g^\star) = O(\mathfrak{R}_n(\mathcal{G})^{2/3}), \tag{80}$$

which implies

$$\mathcal{R}(\hat{g}) - \mathcal{R}(g^\star) = O(n^{-1/3}). \tag{81}$$

*Remark* B.5. The probability $1 - 3\zeta$ follows from a union bound over three high-probability events: one controlling the generalization error in Term I, and two ensuring that the auxiliary estimators $\hat{q}_{\tau+\delta}$ and $\hat{q}_{\tau-\delta}$ remain within the prescribed $L_2$-neighborhoods of $q_{\tau+\delta}$ and $q_{\tau-\delta}$, respectively.

$\square$

## B.4. Proof of Hölder-type Norm Equivalence

**Lemma B.6** (Norm equivalence for Hölder classes). *Let $\mathcal{X} \subset \mathbb{R}^d$ be compact with nonempty interior, and let $P_X$ be a distribution supported on $\mathcal{X}$ whose density is bounded away from zero and infinity. Suppose $h \in C^s(\mathcal{X})$ for some $s \in (0, 1]$ and*

$$\|h\|_{C^s(\mathcal{X})} \leq R.$$

*Then there exists a constant $C_H > 0$, depending only on $(s, d, \mathcal{X}, P_X)$, such that*

$$\|h\|_{L_\infty(\mathcal{X})} \leq C_H R^{\frac{d}{2s+d}} \|h\|_{L_2(P_X)}^{\frac{2s}{2s+d}}. \tag{82}$$

*Proof.* Fix any $x_0 \in \mathcal{X}$. By the Hölder continuity of $h$ and the reverse triangle inequality, for all $x \in \mathcal{X}$, we have $|h(x)| \geq |h(x_0)| - R\|x - x_0\|^s$. Let $B(x_0, r)$ denote the Euclidean ball of radius $r > 0$ centered at $x_0$. Assuming $r$ is small enough such that $|h(x_0)| > Rr^s$, we can lower bound the $L_2$-norm on this local ball:

$$\|h\|_{L_2(P_X)}^2 \geq \int_{B(x_0,r)\cap\mathcal{X}} h(x)^2 \, dP_X(x) \gtrsim r^d \left(|h(x_0)| - Rr^s\right)^2, \tag{83}$$

where the implicit constant depends on the density of $P_X$ and the geometry of $\mathcal{X}$. Taking the square root and rearranging terms to isolate $|h(x_0)|$ yields the bias-variance decomposition:

$$|h(x_0)| \lesssim \underbrace{Rr^s}_{\text{bias}} + \underbrace{\|h\|_{L_2(P_X)}r^{-d/2}}_{\text{variance proxy}}. \tag{84}$$

(Note: If $|h(x_0)| \leq Rr^s$, this inequality holds trivially). We optimize the right-hand side over $r > 0$ by choosing the balancing radius:

$$r \asymp \left( \frac{\|h\|_{L_2(P_X)}}{R} \right)^{\frac{2}{2s+d}}. \tag{85}$$

Substituting (85) back into (84), we obtain

$$|h(x_0)| \lesssim R^{\frac{d}{2s+d}} \|h\|_{L_2(P_X)}^{\frac{2s}{2s+d}}.$$

Since $x_0$ is arbitrary, taking the supremum over $x_0 \in \mathcal{X}$ completes the proof. $\qquad\square$

*Remark* B.7 (Connection to Gagliardo-Nirenberg Inequality). The inequality (82) constitutes a specific instance of the celebrated Gagliardo-Nirenberg interpolation inequality. In its general form on a bounded domain satisfying the cone condition, for indices $1 \leq q, r \leq \infty$ and $j < m$, the inequality states:

$$\|D^j u\|_{L_p} \leq C_1 \|D^m u\|_{L_r}^\theta \|u\|_{L_q}^{1-\theta} + C_2 \|u\|_{L_q}. \tag{86}$$

In our setting, we aim to bound the function value ($j = 0$) in the $L_\infty$-norm ($p = \infty$) using the $L_2$-norm ($q = 2$). The Hölder smoothness constraint $\|\cdot\|_{C^s} \leq R$ serves as the high-order derivative control, corresponding to the limiting case of Sobolev regularity with $m = s$ and $r = \infty$. The interpolation parameter $\theta$ is governed by the dimensional scaling relation:

$$\frac{1}{p} = \frac{j}{d} + \theta \left( \frac{1}{r} - \frac{m}{d} \right) + \frac{1-\theta}{q}$$

$$\implies \quad 0 = 0 + \theta \left( 0 - \frac{s}{d} \right) + \frac{1-\theta}{2}. \tag{87}$$

Solving (87) for $\theta$ yields $\theta = \frac{d}{2s+d}$. Consequently, the exponent for the $L_2$-term is $1 - \theta = \frac{2s}{2s+d}$, which precisely matches the rate in (82).

# C. Supplementary Material

## C.1. Gap between $F_{S|X}(\hat{q}(x))$ and $\pi(x)$

In this part, we supplement the results proved in Plassier et al. (2025a).

**Assumption C.1.** Let $R = S - \hat{q}_\tau(X)$ be the random variable representing the residuals of scores. We assume the densities $f_R$ and $f_{R|X}$ exists for each $x \in \mathcal{X}$, and the likelihood ratio is bounded for each $x \in \mathcal{X}$:

$$\sup_{r \in \mathbb{R}} \frac{f_{R|X=x}(r)}{f_R(r)} \le \Lambda.$$

Here, we adopt a slightly stronger [2] version than (Plassier et al., 2025a) and suppress complex notation to enhance readability. Assumption C.1 essentially characterizes the quality of the quantile estimator, i.e., to what extent the quantile estimator removes the covariate dependency on scores.

**Theorem C.2** (Theorem 2 in Plassier et al. (2025a))**.** *Suppose Assumption C.1 holds. For any target level $\tau$ such that $(1 - \tau) \in ((m+1)^{-1}, 1)$, the deviation between the true and implied coverage is bounded by:*

$$-\Delta^-_{m,\tau} \le \pi(x) - F_{S|X}(\hat{q}_\tau(x)) \le \Delta^+_{m,\tau}, \tag{88}$$

*where the strictly positive slack terms are given by:*

$$\Delta^-_{m,\tau} = \Lambda \left(1 - \frac{k_{cal}}{m+1}\right) [F_R(0)]^{m+1}, \tag{89}$$

$$\Delta^+_{n,\tau} = \Lambda \left(\frac{k_{cal}}{m+1}\right) [1 - F_R(0)]^{m+1}, \tag{90}$$

*with $k_{cal} = \lceil (m+1)\tau \rceil$ being the conformal score rank, and $F_R(0) = \mathbb{P}(S \le \hat{q}_\tau(X))$ being the marginal coverage of the uncalibrated estimator.*

*Remark* C.3. Since we assume the consistency of $\hat{q}_\tau$, we have $F_R(0)$ converges to $\tau$ in probability. Thus, the slack terms decay exponentially with $m$, implying that $F_{S|X}(\hat{q}_\tau(x))$ converges exponentially fast to $\pi(x)$.

We note that the conformalization step is still substantial when $n$ is small (e.g., smaller than 100), while $\Lambda$ may also decrease as the quantile estimator improves (and originally, the size of samples used for training the quantile estimator). Hence, with a reasonably large sample size of $\mathcal{D}_{cal}$, we can focus on characterizing the finite-sample performance of our method by analyzing the MSE of $F_{S|X}(\hat{q}_\tau(x))$, and naturally the surrogate, expected risk of density-weighted pinball loss.

---

[2]We assume the existence of the PDF, while in (Plassier et al., 2025a), the continuity of CDF is assumed.

