# OpenReview forum: "Colorful Pinball: Density-Weighted Quantile Regression for Conditional Guarantee of Conformal Prediction"
_ICML.cc/2026/Conference — ICML 2026 regular_

### Official Review · Reviewer_pp4t · 2026-03-09

**Soundness:** 3
**Presentation:** 2
**Significance:** 3
**Originality:** 3
**Overall Recommendation:** 4
**Confidence:** 3

**Summary:**

The paper tackles to solve the problem with conformal conditional guarantees. They take the local quantile method as the base framework for conformal regression -- which is done through computing a quantile per x, and then adding a conformal residual to everything. Their approach is to train expand the MSCE loss and optimize a local quantile function. If I understood correctly they first train a quantile estimator, then they train fine tune one model with weights defined over the deviation of the quantile estimators, and then the apply conformal prediction on the final residual. So basically the split the main calibration set into a calibration and tuning set, and the train and finetune estimators tuning set.

**Compliance With Llm Reviewing Policy:**

Affirmed.

**Final Justification:**

My score is already at accept. Therefore I maintain it.

**Key Questions For Authors:**

1. All of a sudden I was introduced to the notation $F_{S|X}$ without any description for F, and the subscript (which I assume is the pdf over the score conditional to the point, but not sure if it is the right notation).

2. I am still not sure how bold is assumption 3.2. I think it is more strict than what Prop. 3.1 says.

3. I am confused with Col1 of lines 235 and 236. Is the value in line 236 exactly equal to $1 - \alpha$? If so why are you approximating it?

4. Can the framework be generally defined for any score distribution without going through the first step? Namely there are now UQ methods that return a distribution over the probabilities / scores. Can they be tuned for conditional coverage through your method?

**Limitations:**

I think there are two main limitations (1) the complexity of the algorithm. But this could stem form me not having a clear vision over the whole setup. (2) The algorithm is data heavy; in order to train the quantile estimator the authors may need an extra large calibration set, which is not aligned with the general CP framework being able to fit for limited data scenarios. I would very much appreciate an ablation both playing with the calibration fraction, and the fractions for the three subsets. That could clearly tell more about which part of the process is more data intensive. I can clearly see it is definitely not the last part.

**Strengths And Weaknesses:**

**Strengths**:
- Importantly the paper points at the important problem of conditional coverage, and through empirical results it seems like they are addressing the issue relatively successfully.
- The paper is theoretically grounded; while for the guarantee to hold, it suffices to only discuss the last step, still the first two steps are theoretically justified. Still since the writing is to complex, I could not follow some parts. On the other hand the method is complex
- I am not sure about it but seems like the method does not rely on a continuous single dimensional output from the model and all blocks could be potentially used for classification as well. While I totally understand that the authors chose to aim for regression in general, the algorithm boils down to a local estimator of the quantile which is trained with a tuning set.


**Weaknesses**:
- This is not a major weakness but the method is perfectly tailored for the regression.
- The paper is very hard to follow. As a reader I jumped from one proposition to the other without clearly knowing what is happening under the hood, and why each proposition is stated. Yes they finally lead to the result successfully, but there is no clear image of "why". This becomes more important the moment at page 6 where the reader realizes that the only block needed for theory to work is the third step. This does not deny the theoretical intuition behind why it works better than other approaches, but if that is the point of the story, I would prefer to have an images of the framework that develops gradually. now I only had confusion until page 6.
- This hard-to-read character of the paper is also in some choices of wording as well.
-**Data intensive regime.** Based on the algorithm in the paper the dataset should be sufficiently large for the algorithm to be applicable. Notably this means that the authors surely need do an ablation on the size of the calibration set and the quality of their method. 20% of the dataset is still huge.


**Writing issues.** I also mentioned it in the questions, some notations are used without proper introduction. I would surely appreciate if after some more complicated propositions (like 3.1) authors could provide an intuitive explanation, so that reader can depict what the result mean.
Also I highly suggest to use the approximation notation in Eq. 11, and 12, instead of equality; unless I am mistaking that the authors are using Taylor's *approximation*.

---

> ### Author Rebuttal · Authors · 2026-03-26
>
> We sincerely thank the reviewer for the careful reading and constructive feedback. We address the concerns below.
>
> ### **Focus on regression**
>
> Our paper, like CQR and RCP, is indeed designed for regression, and we will make this positioning more explicit.
>
>
> ### **Logic flow of the paper**
>
> We clarify the logic of the paper as follows, and we will add a sketch sentence in our revision.
>
> - Section 2 explains the motivation of MSCE. Since the bias term converges quickly, the variance of $\pi(X)$ becomes the dominant term in the MSE of $\pi(X)$. Thus, controlling MSCE means controlling this variance, which in turn supports high-probability conditional coverage via Bernstein-type inequality.
> - At the beginning of Section 3, we explain why MSCE cannot be optimized directly: we do not observe the full conditional distribution $Y \mid X$, and therefore cannot evaluate $\pi(X)$ itself. We instead use the result of Plassier et al. (2025a) and take $F_{S \mid X}(\hat q_\tau(x))$ as a surrogate for $\pi(X)$, since their gap can be controlled; see Appendix C.1.
> - Proposition 3.1 is included mainly for completeness and comparison with our own route. It presents the pointwise result of Plassier et al. (2025a), explains why quantile regression arises naturally, and we provide a self-contained proof.
> - We then introduce our own route by designing a tractable objective. MSQE serves as a surrogate for MSCE, and the Taylor expansion leading to Eq. (14) shows that the main term is a density-weighted pinball loss plus constants. We then explain Proposition 3.4 to make its role clearer.
> - Next, in Section 4.1, motivated by limited calibration set size, rather than estimating the conditional density $f_{S \mid X}(q_\tau(X))$ directly, we use Eq. (18) to build a finite-difference estimator, which is substantially lightweight.
>
> ### **Data requirements**
>
> We clarify that the datasets are of moderate size, and the 20% calibration split follows common practice in prior work. All methods are evaluated on the same calibration split (for each random split), which ensures a fair comparison. For completeness, we supplement the experiments on Naval and SGEMM dataset with main methods and the calibration proportion reduced to **5%**. We observe that the advantages of CPCP remain.
>
> **Naval**
>
> | Method             | WSC             | MSCE_30         |
> |:-------------------|:----------------|:----------------|
> | Split              | 0.5736 ± 0.0631 | 0.0349 ± 0.0088 |
> | PLCP       | 0.7236 ± 0.0891 | 0.0151 ± 0.0082 |
> | GS   | 0.7031 ± 0.0567 | 0.0168 ± 0.0048 |
> | CQR        | 0.7122 ± 0.0809 | 0.0186 ± 0.0110 |
> | RCP        | 0.7834 ± 0.0624 | 0.0086 ± 0.0031 |
> | CPCP-Clip+Mix-0.02 | 0.8327 ± 0.0668 | 0.0074 ± 0.0025 |
>
> **SGEMM**
>
> | Method             | WSC             | MSCE_30         |
> |:-------------------|:----------------|:----------------|
> | Split              | 0.7550 ± 0.0161 | 0.0052 ± 0.0006 |
> | PLCP       | 0.7779 ± 0.0386 | 0.0041 ± 0.0014 |
> | GS   | 0.8025 ± 0.0652 | 0.0034 ± 0.0022 |
> | CQR        | 0.8413 ± 0.0225 | 0.0018 ± 0.0010 |
> | RCP       | 0.8506 ± 0.0176 | 0.0015 ± 0.0004 |
> | CPCP-Clip+Mix-0.02 | 0.8598 ± 0.0134 | 0.0013 ± 0.0004 |
>
>
> Moreover, we clarify that **conditional guarantees are inherently harder than marginal guarantees**. Even for marginal coverage, Eq. (1) shows that one already needs a reasonably large sample size to make the bias small (e.g., 100 samples for 1% marginal coverage bias). For conditional guarantees, the needed sample size should be much larger.
>
> On the other hand, our method has adopted a relatively constrained strategy: we estimate three conditional quantiles rather than a full conditional density. For this reason, we believe the data requirement is moderate rather than excessive.
>
>
>
> ### **Taylor expansion**
>
> Our derivation in Proposition 3.4 uses the Lagrange form of the remainder, so Eq. (11) and Eq. (12) are exact identities rather than approximations. The lower-order terms together with the remainder equal the original quantity exactly.
>
> ### **Questions**
>
> 1. $F_{S \mid X}$ is the conditional CDF of the nonconformity score given $X$. It is first introduced in Section 1, line 50 of the right column.
>
> 2. Assumption 3.2 is not comparable to Prop 3.1. It is used only to illustrate the leading term in Proposition 3.4 and induce our surrogate. It is not required for our main theoretical results, so we consider to replace it with a descriptive statement in revision.
>
> 3. $f_{S \mid X}$ is the conditional density of the score, first introduced in Section 1, line 53 of the right column and line 72 of the left column. We have $F_{S \mid X}(q_\tau(x)) = \tau$, not $f_{S \mid X}(q_\tau(x)) = \tau$, and this identity is exactly what leads to Eq. (18).
>
> 4. We agree that one could directly tune a conditional distribution to minimize MSCE. Our point is that estimating a full conditional distribution is much harder than estimating conditional quantiles, especially when the calibration set is limited.

---

> > ### Author Rebuttal · Reviewer_pp4t · 2026-04-03
> >
> > Many thanks for answering my questions. Since I am already assigning an accepting score, I will maintain it.

---

### Official Review · Reviewer_LVdu · 2026-03-12

**Soundness:** 2
**Presentation:** 2
**Significance:** 2
**Originality:** 2
**Overall Recommendation:** 3
**Confidence:** 4

**Summary:**

This paper focuses on a problem in Conformal Prediction area,how to improve conditional coverage in the case of finite samples. The article suggests that there is a deviation between the pinball loss and the conditional coverage target. This paper modifies the representation of loss based on Taylor expansion,constructs a new computational framework called Colorful Pinball Conformal Prediction (CPCP) and tests the effectiveness of CPCP on some regression datasets.

**Compliance With Llm Reviewing Policy:**

Affirmed.

**Key Questions For Authors:**

1 The paper itself stated at the beginning “As machine learning systems are increasingly deployed in high-stakes environments，”, but the dataset is relatively ordinary regression prediction, such as Diamond value prediction. Can we try using datasets related to autonomous driving or medical surgery.

2 The algorithm incorporates weight clipping and loss mixing in the practical process, which is not covered in the theoretical section. Can the author theoretically discuss or experimentally examine how these two steps affect the results.

**Limitations:**

There are several strong assumptions in the article, like “Local Hölder norm equivalence”，“q isthree times continuously differentiable in τ”，In some data scenarios, it is necessary to test whether these hypotheses hold true.

**Strengths And Weaknesses:**

Strengths

1The method is in the category of weighted quantile regression, but due to its computation steps, especially "the conditional density of scores evaluated at the true quantile", the weights come from the conditional density of scores and differs from traditional weighted quantile regression, so its originality is good.

2 The overall presentation is clear and the supplementary materials are abundant.

Weakness
1The paper itself stated at the beginning “As machine learning systems are increasingly deployed in high-stakes environments”, in high-risk scenarios, stronger reliability is required, but experiments mainly focus on traditional regression prediction tasks such as value prediction, without actual testing in high-risk tasks such as autonomous driving, healthcare, and so on.

---

> ### Author Rebuttal · Authors · 2026-03-25
>
> We sincerely thank the reviewer for efforts in reviewing our work. We address the concerns below.
>
> ### **Discussion of high-stakes scenarios**
>
> We would like to clarify that the discussion in the introduction is meant to motivate the broader importance of **uncertainty quantification** in machine learning, especially in high-stakes domains such as healthcare and finance. Our goal is to explain why reliable uncertainty estimates matter, rather than to position this paper as an application paper for a specific high-stakes task.
>
> More importantly, conformal prediction is fundamentally developed for settings with a **fixed and well-defined label space**, where the goal is to construct prediction sets or intervals with distribution-free coverage guarantees. For this reason, the standard evaluation protocol in the conformal prediction literature is based on classical regression or classification benchmarks, as in representative prior works such as CQR, PLCP and RCP. We therefore believe that standard regression tasks are appropriate for demonstrating the merits of a **methodological contribution**.
>
> While there are application-oriented conformal works for more complex settings such as image segmentation, such extensions usually require substantial task-specific reformulation. In contrast, our paper focuses on core methodology. Problems such as autonomous driving or medical decision-making often involve structured outputs or sequential decisions, so adapting our framework to those settings would require significant new technical development and is beyond the scope of this paper.
>
> ### **Theoretical motivation for weight clipping and loss mixing**
>
> Weight clipping and loss mixing are **not ad hoc additions**. As stated at the end of Section 4.2, they are practical modifications motivated by the theoretical bottlenecks identified in the **proof** of our main theorem.
>
> Specifically, in Eq. (68), we need a **uniform lower bound** on  $\hat\Delta_{\delta}(x)$, which appears in the denominator when controlling the weight estimation error term $||\hat{w} - w_\delta||_{2,n}$ (see Eq. (70)). Establishing such a bound requires the Hölder-type norm equivalence assumption, which is inherently **instance-dependent**. Weight clipping is therefore introduced to improve robustness in practice by preventing instability when the estimated denominator becomes too small.
>
> From the algorithmic side, **loss mixing** plays a complementary role. When some estimated weights are extremely small, the corresponding samples may contribute almost nothing to optimization. Loss mixing preserves their influence to some extent, trading a small amount of bias for improved stability and a healthier effective sample size, especially when the calibration set is limited.
>
> We agree that this connection could be explained more clearly, and we are happy to make it more explicit in the revision.
>
> ### **Validation of assumptions**
>
> We first clarify that these assumptions are standard regularity conditions used to characterize **how doable the problem is**. At the same time, many of them are **instance-dependent**, and fully verifying them in practice is generally unrealistic. This is common in machine learning theory, where assumptions are introduced to make the analysis tractable and to characterize when a method is expected to perform well. Without such regularity assumptions, the theory is often too general to be meaningful, as it must also cover highly irregular cases.
>
> For the local norm equivalence assumption, its plausibility depends on both the regularity of the quantile regressor class and realizability. For the former, we provide the Hölder class as a sufficient example, which can cover common smooth function classes such as RKHSs with smooth kernels, and in some settings neural networks with appropriate activations. For the latter, realizability means that the true conditional quantile function lies in the chosen function class. This is a standard theoretical assumption, but it is generally not directly verifiable in practice.
>
> Similarly, the assumption that the conditional quantile function $q_\tau(x)$ is three times continuously differentiable is also instance-dependent. It may fail when the conditional law $Y \mid X=x$ is highly irregular. More precisely, this is a smoothness assumption on the conditional distribution: it is typically justified when the conditional density is sufficiently smooth and strictly positive around the target quantile. Such conditions often hold for well-behaved continuous models, including many continuous exponential-family distributions. In practice, one can only judge whether such assumptions are plausible based on domain knowledge; they are not usually conditions that can be rigorously verified from finite data.
>
> We will clarify this point in the revision and emphasize that these assumptions are mainly used to support the theoretical analysis, rather than as directly testable requirements in applications.

---

> > ### Author Rebuttal · Reviewer_LVdu · 2026-04-05
> >
> > Resolved

---

> > > ### Author Response · Authors · 2026-04-06
> > >
> > > We appreciate your confirmation that our previous response has fully resolved your concerns. In light of this updated assessment, we would be grateful if you could consider raising your score accordingly, especially given the initial negative score assigned with high confidence (4).
> > >
> > > To provide broader context on motivation to this work, we emphasize the critical distinction between marginal and conditional coverage in conformal prediction (Barber et al., 2021). Basic marginal coverage only provides an average guarantee; it inevitably suffers from spatial imbalances where severe under-coverage in complex, high-risk regions is artificially compensated by over-coverage in simpler regions (Romano et al., 2019, Braun et al. 2025a). This flaw introduces significant downstream risks, as the predictive uncertainty is **not reliable for certain specific instances at hand**. Moving beyond classical fairness evaluations, achieving true conditional coverage for *each individual input instance* is essential for safe deployment. By directly optimizing this, our work addresses a core bottleneck in the field.
> > >
> > > For completeness, we briefly reiterate the core novelty and contributions of our work:
> > >
> > > - We identify the limitations of naive quantile regression for conditional coverage and develop a substantially sharper approximation to the MSCE.
> > > - We propose a principled algorithmic framework for directly minimizing the MSCE, in contrast to many existing works that only target relaxed conditional coverage $P(Y\in\mathcal{C}_\alpha(X)\mid H(X))$, while incorporating carefully designed mechanisms to ensure practical robustness.
> > > - We establish a non-asymptotic generalization theory with estimated reciprocal weights, which in turn yields control of the MSCE.
> > > - We validate our approach through comprehensive experiments and extensive ablations, including detailed studies of each major algorithmic component.

---

### Official Review · Reviewer_954C · 2026-03-13

**Soundness:** 3
**Presentation:** 3
**Significance:** 3
**Originality:** 3
**Overall Recommendation:** 4
**Confidence:** 3

**Summary:**

Methodologically, the paper proposes a three-stage framework, CPCP, built on rectified conformal prediction to improve exact conditional coverage. A central idea is a Taylor expansion argument that yields a sharper approximation of MSCE. It relates the conditional coverage gap to a density-weighted pinball loss, thereby motivating the proposed method. Theoretically, the paper establishes non-asymptotic bounds and rate of convergence via a Rademacher complexity analysis. Empirically, the method demonstrates descent performance on both worst-slab conditional coverage and MSCE.

**Compliance With Llm Reviewing Policy:**

Affirmed.

**Final Justification:**

Overall, I maintain my positive evaluation of the paper. It is theoretically grounded and makes a meaningful contribution to the literature on conditional conformal prediction.

**Key Questions For Authors:**

Could the authors provide more intuition on how the quality of the estimated weights affects the method’s performance, and include some numerical results to illustrate this sensitivity?

**Limitations:**

Yes

**Strengths And Weaknesses:**

**Strength.** The paper is well motivated and has a very clear problem set up. Unlike prior work that mainly focus on group-wise coverage, the paper considers an exact notion of conditional coverage, which I find both meaningful and relatively novel. The Taylor-expansion based derivation is a meaningful insight. The overall three-stage CPCP pipeline is also clearly designed and supported by the empirical results.

**Weaknesses.**
* A clearer comparison of runtime and computational cost against the baselines would also help assess the practical tradeoff of the proposed method.
* The paper would benefit from an ablation and sensitivity analysis, especially in settings where the weight estimates are noisy or misspecified. It would be valuable to assess how sensitive the proposed method is wrt weight estimation error and how this impacts final performance.

---

> ### Author Rebuttal · Authors · 2026-03-26
>
> We sincerely thank the reviewer for the careful reading of our paper and for the positive assessment of its motivation, novelty, and empirical results. We address the concerns below.
>
> ### **Computational cost**
>
> In our setting, computational cost is not a major concern. CPCP only uses the calibration set, which typically contains a few hundred to a few thousand samples, and a simple 3-layer MLP is sufficient for the quantile regressors. More broadly, this is usually not a major issue for **split conformal** methods. In contrast, the main computational burden arises in **full conformal** procedures, such as Gibbs et al. (2025a), which require repeatedly running conformal inference, and often quantile regression as well, over many candidate values $y_{\mathrm{test}} \in \mathcal{Y}$.
>
> To provide a concrete reference, we report the average runtime on SGEMM on our device under a 60\%-20\%-20\% train/test/calibration split:
>
> | Method   | Average runtime |
> |----------|-----------------|
> | Split CP | 16.6s           |
> | PLCP     | 18.6s           |
> | Gaussian Scoring       | 40.5s           |
> | CQR      | 17.4s           |
> | RCP      | 18.6s           |
> | CPCP     | 25.6s           |
>
> These results suggest that CPCP is somewhat slower than standard split conformal baselines, but remains in the same practical range.
>
> ### **Sensitivity to weight estimation**
>
> Due to space limitations in the main paper, we report several ablation results in the appendix, including Appendix A.5. For weight estimation, we study three quantile gaps, corresponding to $\delta \in \lbrace 0.01, 0.02, 0.05 \rbrace$. Given a fixed calibration set and function class, $\delta$ directly affects the quality of the estimated weights.
>
> Intuitively, when calibration data are limited, choosing $\delta = 0.01$ can be unstable, since the sample may not support reliably distinguishing the 89\% and 91\% quantiles; in that case, a larger choice such as $\delta = 0.05$ is more stable. Empirically, however, we find that CPCP is not very sensitive to $\delta$. More importantly, across all three choices of $\delta$, CPCP consistently shows clear advantages over the baselines in conditional coverage metrics; see Tables 4--8.
>
> We agree that this point is important, and we are happy to make the sensitivity discussion more explicit in the revision, space permitting.

---

> > ### Author Rebuttal · Reviewer_954C · 2026-04-02
> >
> > I thank the authors for their response and further clarification. I will maintain my score.

---

### Decision · Program_Chairs · 2026-04-30

**Decision:**

Accept (regular)

**Comment:**

The paper proposes a density-weighted quantile regression framework that improves conditional coverage in conformal prediction via a Taylor-expansion–motivated weighted pinball loss. The submission received three final reviews: two weak accepts and one weak reject, with no score changes after rebuttal, although one reviewer indicated their concerns were fully resolved.

Reviewers agreed on several strengths: the paper tackles an important problem, provides a well-motivated theoretical contribution with non-asymptotic guarantees, and demonstrates consistent empirical improvements. The approach is viewed as technically sound and relevant to the field.

Concerns mainly relate to clarity of presentation, methodological complexity, and missing empirical analyses (e.g., sensitivity and cost). However, these issues do not affect the correctness or core contribution, and several were addressed in the rebuttal.

Overall, the paper makes a solid and meaningful contribution to conditional conformal prediction. Therefore, I recommend acceptance.